# Recent Advancements in Lateral Flow Assays for Food Mycotoxin Detection: A Review of Nanoparticle-Based Methods and Innovations

**DOI:** 10.3390/toxins17070348

**Published:** 2025-07-11

**Authors:** Gayathree Thenuwara, Perveen Akhtar, Bilal Javed, Baljit Singh, Hugh J. Byrne, Furong Tian

**Affiliations:** 1School of Food Science and Environmental Health, Technological University Dublin, Grangegorman, D07 ADY7 Dublin, Ireland; d24127925@mytudublin.ie (G.T.); d24127942@mytudublin.ie (P.A.); baljit.singh@tudublin.ie (B.S.); 2Nanolab Research Centre, Physical to Life Sciences Research Hub, Technological University Dublin, Camden Row, D08 CKP1 Dublin, Ireland; hugh.byrne@tudublin.ie; 3MiCRA Biodiagnostics Technology Gateway and Health, Engineering & Materials Science (HEMS) Research Hub, Technological University Dublin, D24 FKT9 Dublin, Ireland

**Keywords:** mycotoxins, lateral flow assay, nanoparticles, singular detection, multiplex detection, food safety

## Abstract

Mycotoxins are responsible for a multitude of diseases in both humans and animals, resulting in significant medical and economic burdens worldwide. Conventional detection methods, such as enzyme-linked immunosorbent assay (ELISA), high-performance liquid chromatography (HPLC), and liquid chromatography-tandem mass spectrometry (LC-MS/MS), are highly effective, but they are generally confined to laboratory settings. Consequently, there is a growing demand for point-of-care testing (POCT) solutions that are rapid, sensitive, portable, and cost-effective. Lateral flow assays (LFAs) are a pivotal technology in POCT due to their simplicity, rapidity, and ease of use. This review synthesizes data from 78 peer-reviewed studies published between 2015 and 2024, evaluating advances in nanoparticle-based LFAs for detection of singular or multiplex mycotoxin types. Gold nanoparticles (AuNPs) remain the most widely used, due to their favorable optical and surface chemistry; however, significant progress has also been made with silver nanoparticles (AgNPs), magnetic nanoparticles, quantum dots (QDs), nanozymes, and hybrid nanostructures. The integration of multifunctional nanomaterials has enhanced assay sensitivity, specificity, and operational usability, with innovations including smartphone-based readers, signal amplification strategies, and supplementary technologies such as surface-enhanced Raman spectroscopy (SERS). While most singular LFAs achieved moderate sensitivity (0.001–1 ng/mL), only 6% reached ultra-sensitive detection (<0.001 ng/mL), and no significant improvement was evident over time (ρ = −0.162, *p* = 0.261). In contrast, multiplex assays demonstrated clear performance gains post-2022 (ρ = −0.357, *p* = 0.0008), largely driven by system-level optimization and advanced nanomaterials. Importantly, the type of sample matrix (e.g., cereals, dairy, feed) did not significantly influence the analytical sensitivity of singular or multiplex lateral LFAs (Kruskal–Wallis *p* > 0.05), confirming the matrix-independence of these optimized platforms. While analytical challenges remain for complex targets like fumonisins and deoxynivalenol (DON), ongoing innovations in signal amplification, biorecognition chemistry, and assay standardization are driving LFAs toward becoming reliable, ultra-sensitive, and field-deployable platforms for high-throughput mycotoxin screening in global food safety surveillance.

## 1. Introduction

Mycotoxins are naturally occurring secondary metabolites produced by various toxigenic fungi. They have the ability to contaminate the entire food chain, starting from agricultural crops and ending up on consumers’ plates [1,2]. Fungal genera such as *Aspergillus*, *Alternaria*, *Fusarium*, *Penicillium*, *Claviceps* and *Stachybotrys* are the primary producers of mycotoxins [1,2,3,4]. These toxigenic fungi can simultaneously produce different mycotoxins in food commodities. Furthermore, food can be contaminated by multiple species of fungi that cannot be fully eliminated, even with good agricultural and manufacturing practices [5,6,7].

To date, more than 500 mycotoxins have been documented [3] and ongoing research suggests that this number has not yet been finalized. Notable mycotoxins include aflatoxins B1, B2, G1, and G2 (AFB1, AFB2, AFG1, AFG2), Ochratoxin A (OTA), various trichothecenes such as deoxynivalenol (DON) and nivalenol (NIV), fumonisins (FB1, FB2, FB3), zearalenone (ZEN), citrinin (CIT), patulin (PAT), cyclopiazonic acid (CPA), tenuazonic acid, and ergot alkaloids (EAs) [5,8,9,10]. These mycotoxins are commonly found in agricultural and food products of both plant and animal origin, posing significant health risks to humans through dietary exposure [11,12,13].

Aflatoxins are among the most toxic mycotoxins and are produced by specific molds, such as *Aspergillus flavus* and *Aspergillus parasiticus*, which thrive in soil, decaying plant matter, hay, and various grains. These toxins commonly contaminate cereals, oilseeds, spices, and tree nuts [14,15]. Aflatoxin M1 (AFM1) may also be present in milk from animals fed contaminated feed [16]. Acute exposure can cause fatal aflatoxicosis due to liver damage [14,17,18,19], while chronic exposure is genotoxic and linked to liver cancer in humans [20].

Ochratoxin A (OTA), produced by *Aspergillus* and *Penicillium species*, frequently contaminates cereals, coffee, dried fruits, wine, grape juice, spices, and liquorice, especially under poor storage conditions [14,21]. OTA primarily causes renal toxicity and may affect fetal development and immune function [22].

Fusarium species, widespread in soil, produce diverse mycotoxins including trichothecenes (DON, NIV, T-2, HT-2), zearalenone (ZEN), and fumonisins [1,23]. These are commonly found in cereals such as DON and ZEN in wheat, T-2 and HT-2 in oats, and fumonisins in maize [1,23,24]. Trichothecenes cause acute gastrointestinal symptoms and chronic immunosuppression [25]. ZEN has estrogenic effects and may impair fertility [26,27]. Fumonisins are associated with liver and kidney toxicity in animals and esophageal cancer in humans [27,28].

According to estimates from the Food and Agriculture Organization (FAO) of the United Nations, approximately 25% of global food crop production is impacted by mycotoxin contamination [29]. This highlights the critical need for effective strategies to address the adverse effects of fungal toxins on food safety and security worldwide. To safeguard human and animal health, regulatory bodies such as the World Health Organization (WHO), FAO, Codex Alimentarius, the European Union (EU), and the United States Food and Drug Administration (FDA) have established strict maximum allowable limits for mycotoxins in food and feed. These thresholds typically fall in the microgram per kilogram (µg/kg) or parts per million (ppm) range [30,31,32,33], reflecting the high toxicity of mycotoxins and the need for rigorous monitoring. Importantly, these values serve as benchmark targets for detection technologies, particularly in evaluating the LOD required for practical deployment.

The EU, through Commission Regulation (EU) 2023/915, has specified detailed maximum allowable levels for various mycotoxins based on food type and intended use. For instance, aflatoxin B1 is limited to 2 µg/kg in groundnuts for direct consumption and 8 µg/kg in groundnuts subjected to physical treatment. Total aflatoxins (sum of B1, B2, G1, and G2) are capped at 4 µg/kg and 15 µg/kg for the respective categories. OTA has limits of 5 µg/kg in unprocessed cereals and 3 µg/kg in processed cereal products. DON is capped at 1250 µg/kg in unprocessed cereals and 200 µg/kg in baby food. ZEN levels are set at 100 µg/kg in unprocessed cereals and 20 µg/kg in baby foods. Fumonisins (B1 and B2) are limited to 2000 µg/kg in unprocessed maize, 1000 µg/kg in maize flour, and 200 µg/kg in maize-derived baby food [31].

In the United States, the FDA has set advisory levels for DON at 1000 µg/kg in finished wheat products such as flour and bran. For fumonisins, allowable limits vary by product type: 2 ppm for degermed dry-milled corn with less than 2.25% fat, and 4 ppm for whole or partially degermed corn products and corn bran. Cleaned corn for masa production (i.e., traditional nixtamalized corn dough used in tortillas and tamales) and popcorn are limited to 4 ppm and 3 ppm, respectively [32].

At the international level, the Codex Alimentarius Commission, advised by the Joint FAO/WHO Expert Committee on Food Additives (JECFA), provides harmonized standards for acceptable mycotoxin levels in food to facilitate global trade and protect consumer health. These values, established through rigorous risk assessments, emphasize minimizing dietary exposure to mycotoxins [33].

Ensuring compliance with these regulations is a collective responsibility shared among agricultural producers, food processors, retailers, and consumers. Adherence to Good Agricultural Practices (GAPs), Good Manufacturing Practices (GMPs), and proper storage protocols is essential to safeguarding food and feed supplies [30]. Globally regulated maximum permissible levels of mycotoxins necessitate continuous monitoring across various commodities to ensure food safety and protect consumer health [30,31,32,33]. Proactive measures are crucial for addressing these challenges and maintaining public health amid changing environmental and agricultural conditions [30,31,32,33].

Accurate and timely detection of mycotoxins is essential for managing the health risks associated with food contamination. Effective detection allows for early intervention, helping to prevent widespread outbreaks and reducing public health impacts. By identifying mycotoxins early, targeted measures can be applied throughout the food supply chain, from production through consumption, thereby minimizing exposure and mitigating adverse health effects. As a result, a range of detection technologies have been developed to identify even trace amounts of mycotoxins in food products, ensuring high sensitivity and specificity in mycotoxin analysis. Techniques such as high-performance liquid chromatography (HPLC) [34,35,36], gas chromatography (GC) [36,37,38,39], enzyme-linked immunosorbent assay (ELISA) [40,41], liquid chromatography–mass spectrometry (LC-MS) [42,43], thin layer chromatography (TLC) [44], polymerase chain reaction (PCR) [30,45,46,47], and various biosensors [48,49,50] offer distinct benefits. These methods provide high-throughput analysis, real-time monitoring, and the ability to test for multiple mycotoxins simultaneously [46].

Although the aforementioned approaches have their advantages, they may not be appropriate for all contexts. Chromatographic methods necessitate the expertise of trained personnel, as well as costly and high-maintenance equipment, high power consumption, and high operational pressure [51]. Similarly, ELISA methods have their limitations, including the requirement for multiple washing steps, non-specific binding, expensive equipment, and large sample volumes. Additionally, the sensitivity and stability of ELISA methods may be suboptimal [52]. While PCR offers high accuracy and sensitivity in detecting mycotoxins, it is labor-intensive and time-consuming [45,46,47].

The quest for rapid, cost-effective, and accurate detection methods continues, as traditional techniques are expensive and can sometimes yield false positives or negatives. Recent advancements in detection technologies, such as microfluidics, lab-on-a-chip systems, smart nanospectroscopy, and sensor technologies, offer promising alternatives for rapid and cost-effective mycotoxin detection [53,54,55,56,57]. However, each method has its drawbacks. Microfluidics can be complex and expensive to fabricate and managing heat can affect accuracy [57]. Lab-on-a-chip systems face challenges with integration, high initial costs, and maintenance [58]. Smart nanospectroscopy is costly and requires specialized training. Sensor technologies often need frequent calibration and can be sensitive to environmental conditions [59].

Given the limitations of existing methods, there is a pressing need for a mycotoxin detection tool that is rapid, precise, sensitive, easy to use, and cost-effective. Lateral flow assay (LFA) devices offer a feasible alternative to conventional instrumental techniques and advanced biosensors. As paper-based biosensors, LFAs can identify various mycotoxins in different food samples within 5 to 30 min [60] and can be optimized for single [60] or multiplex [60] detection. The assay utilizes capillary action within the nitrocellulose membrane to streamline the testing process (Figure 1), while its fibrous structure helps eliminate impurities [61]. LFAs are stable under varying environmental conditions, have a long shelf life, and require minimal sample input without extensive preparation or washing steps [62]. Designed for simplicity, they require little specialized expertise [63], can be integrated with electronic devices, and hold strong commercialization potential [63]. These features make LFAs suitable for food mycotoxin detection beyond laboratory settings. Their speed, affordability, and field deployability have attracted considerable attention, particularly in resource-limited and remote communities where trained personnel may be scarce. In such settings, LFAs are especially valuable due to their portability [63].

However, to evaluate the regulatory compliance and real-world applicability of LFA devices, it is crucial to contextualize their sensitivity in relation to internationally accepted mycotoxin thresholds. Many LFA studies report limit of detection (LOD) in nanograms per millilitre (ng/mL), particularly when using aqueous-based extracts. In contrast, legal safety thresholds established by global regulatory agencies such as the FAO, Codex Alimentarius, the EU, and the FDA are typically expressed in micrograms per kilogram (µg/kg) or parts per million (ppm) [31,32,33]. To harmonize these units, an approximate conversion of 1 ng/mL ≈ 1 µg/kg is often assumed for liquid matrices. However, this equivalence is heavily influenced by the sample matrix’s density, composition, and extraction protocol. Since most food samples are initially solid, they undergo solvent-based extraction prior to analysis. This step inherently dilutes the analyte concentration, potentially leading to lower LODs reported in ng/mL than would be present in the original solid matrix. As a result, LODs expressed in liquid-phase units may overestimate the assay’s sensitivity relative to the actual mycotoxin burden in foods. Therefore, care must be taken when comparing experimental LODs with statutory safety limits, and matrix-corrected unit conversions are essential to accurately determine whether LFA platforms meet the sensitivity requirements necessary for regulatory compliance and effective food safety surveillance.

Recent advancements in LFAs have notably improved their sensitivity, specificity, usability, and multiplexing capabilities for mycotoxin detection, establishing them as critical tools for food safety. These improvements are achieved through several strategies, focusing on assay kinetics and signal amplification. Signal amplification methods enhance the performance of LFAs by increasing the colorimetric contrast. The integration of nanomaterials such as gold nanoparticles (AuNPs) [64,65,66], fluorescent nanoparticles (FNs) [67,68,69,70,71], magnetic nanoparticles [72,73,74], carbon nanoparticles (CNPs) [75,76], and carbon nanotubes (CNTs) [77,78] has significantly advanced LFA performance (Figure 2A. AuNPs are particularly favored for their compatibility and visibility, while FNs, including quantum dots (QDs) and upconverting nanoparticles (UCNPs), offer superior sensitivity for low-concentration targets [79,80,81,82,83,84]. Chemical enhancement through enzymatic reactions, such as those catalyzed by horseradish peroxidase (HRP), amplifies color signals, while nanozymes, synthetic enzymes with high catalytic stability, offer improvements in sensitivity and stability [79,80,81,82,83,84] (Figure 2A). Additional methods like silver enhancement and double gold conjugation significantly boost signal intensity, achieving sensitivity gains of up to 1000-fold compared to traditional methods [85,86,87]. The incorporation of readers into LFA systems enables the detection of signals not easily visible by the naked eye, using external stimuli to amplify and quantify signals, with various readers including fluorescence [88,89], surface-enhanced Raman scattering (SERS) [90,91], photothermal [92,93], electrochemistry [94], magnetic amplification [95,96] and smartphone-based readers offering different advantages [97] (Figure 2B).

The kinetics of transport of analytes and nanoparticle-antibody conjugates through the capillary flow on the LFA membrane and reaction are critical for the sensitivity of the immunochromatographic assay. Transport kinetics, influenced by convection and diffusion, determine how rapidly molecules move through the assay membrane [98,99,100]. High convection rates enable faster molecular movement, while diffusion, which is slower, affects the spreading of molecules. Reaction kinetics, concerning the rate of reactions at the assay surface, impact sensitivity; a higher reaction rate improves sensitivity, whereas a lower rate constrains it [98,100]. Enhancing sensitivity involves optimizing reaction rates and reactant concentrations to improve interactions at the assay surface [98,100]. Techniques such as using sequential flow [101], concentrating reactants, before introduction to the assay, such as through magnetic separation [102,103] (illustrated schematically in Figure 3) and increasing the number of binding sites can significantly enhance LFA performance [100,104,105,106,107,108,109].

Figure 3 illustrates a sample enrichment process designed to decrease the LOD in LFAs. It depicts the addition of magnetic nanoparticles conjugated with specific antibodies to a food sample containing the target mycotoxin. The illustration also shows the application of an external magnetic field to capture and concentrate the mycotoxin-antibody-nanoparticle complexes from the sample. This preconcentration step improves the sensitivity of the LFA by reducing matrix interference and increasing the analyte concentration. The diagram highlights how this enrichment strategy enhances detection accuracy and lowers the LOD for mycotoxin analysis in complex food matrices.

Improving specificity is essential for effective mycotoxin detection. This involves minimizing nonspecific binding (NSB) and using highly selective affinity molecules. NSB can result from substances in the sample binding nonspecifically to assay components, interactions between conjugated labels and capture antibodies or membranes, or physical trapping of label aggregates [98,100,110,111,112]. Optimizing assay components through surface modification and blocking of labels, adjusting label size and concentration, and selecting appropriate running buffer composition can reduce NSB while enhancing specific binding [100,110,113,114,115,116]. The pore size of membranes also influences assay performance by affecting flow dynamics and nonspecific binding. Smaller pores can slow sample flow, improving sensitivity but potentially extending assay time, while larger pores facilitate faster flow but may increase background noise (Figure 4) [98,100].

The choice of affinity molecules greatly impacts assay specificity. Antibodies, though traditional and widely used, can suffer from variability and cross-reactivity [116,117]. Aptamers, as oligonucleotides, offer high stability, lower variability, and the ability to bind non-immunogenic targets, making them promising alternatives [118,119,120]. However, commercial use of aptamers is still developing, and antibodies remain preferred for their ability to detect larger targets [118,119,120,121] (Figure 4).

Figure 4 illustrates the key factors influencing the optimization of LFAs. It shows how membrane pore sizes affect flow dynamics and analyte capture efficiency, with smaller pores typically enhancing sensitivity by providing a more controlled and efficient analyte transfer. It also illustrates the impact of selecting different affinity molecules, such as antibodies and aptamers, on the assay’s specificity and sensitivity, emphasizing the need for precise binding interactions to minimize cross-reactivity and maximize detection accuracy. Additionally, it also highlights the role of buffer conditions, including variations in pH, ionic strength, and the presence of stabilizers, in optimizing assay performance. Proper buffer conditions are crucial for maintaining the stability and functionality of the affinity molecules, ensuring optimal binding and consistent assay results. This comprehensive diagram demonstrates how careful optimization of these factors collectively improves the specificity, sensitivity, and overall performance of LFAs by enhancing analyte binding, reducing non-specific interactions, and maintaining reliable flow dynamics.

Despite the various techniques available to improve the sensitivity and specificity of LFAs, significant challenges remain for their widespread adoption and optimization for mycotoxin detection in food. Achieving the required sensitivity to detect mycotoxins at extremely low concentrations, often in the parts per billion (ppb) or even trillion (ppt) levels, is a primary obstacle. Ensuring high specificity to avoid cross-reactivity with other substances in complex food matrices is also critical and challenging, as interference from food matrices can affect the accuracy and reliability of LFA results, complicating interpretation.

Additional challenges include maintaining the stability and self-life of LFAs under varying storage conditions, standardizing protocols across different mycotoxins and food types, and ensuring affordability and accessibility. Regulatory approval for new LFA technologies and the integration of digital solutions for enhanced data management also pose significant hurdles. Addressing these challenges is essential for enhancing the reliability, efficiency, and widespread use of LFAs in safeguarding food safety against mycotoxins.

This paper provides an in-depth review of recent advancements in LFA technology for detecting mycotoxins in food products. It evaluates the impact of various nanoparticle materials including gold, silver, fluorescent, magnetic, and carbon-based nanoparticles on key assay performance metrics such as sensitivity, specificity, and LOD. The review also highlights assay design, multiplexing capabilities, and the integration of portable and mobile readout systems. These developments have established LFAs as highly effective tools for rapid, accurate, and cost-efficient mycotoxin detection, which is crucial for ensuring food safety and public health. Addressing ongoing challenges and refining LFA technologies will be essential for enhancing their practical application in detecting foodborne mycotoxin.

## 2. Research Method

A comprehensive literature search was conducted across major scientific databases, including PubMed, Web of Science, Science Direct, and Google Scholar. The search strategy incorporated a range of keywords and phrases to capture a broad spectrum of relevant studies. These keywords included “mycotoxins”, “lateral flow assay”, “LFAs”, “nanoparticle-based detection”, “gold nanoparticles”, “quantum dots”, “metal–organic frameworks”, “upconverting nanoparticles”, “enzymes”, “sensitivity”, “food safety”, “aflatoxins”, “zearalenone”, “deoxynivalenol”, “fumonisins”, and “ochratoxin”. This extensive search was designed to encompass both foundational research and the most recent advancements in the field of nanoparticle-based LFAs.

The selection of studies was guided by specific inclusion and exclusion criteria to ensure the relevance and quality of the included research. Eligible studies were original research articles, or significant case studies that focused on LFAs for detecting mycotoxins in food and feed samples. These studies reported advancements in nanoparticle technology, including innovations in nanoparticle synthesis, signal amplification techniques, or improvements in assay sensitivity. The publication period was limited to January 2015 to December 2024, and only studies published in English were included. Studies that did not focus on mycotoxin detection, did not utilize LFAs, addressed non-food applications of nanoparticle-based detection, or did not present novel advancements in LFA technology, were excluded.

Data extraction was precisely conducted to ensure consistency and accuracy. Key data extracted from each study included the type of sample or matrix, target analyte, type of nanoparticles used, reported technological advancements, detection limits, year of publication, and citation details. Each study was then critically assessed for methodological rigor and the relevance of its contributions to advancements in nanoparticle-based LFA technology. This evaluation focused on the novelty of reported innovations, robustness of experimental designs, and overall impact on mycotoxin detection.

A structured data analysis was performed to evaluate advancements in nanoparticle-based LFAs. Data analysis included several visualization techniques to comprehensively assess the advancements. A total of 78 studies were included in the review. A comprehensive data catalogue (Table 1) was compiled, organizing the extracted information by sample type, target analyte, nanoparticle type, technological advancements, sensitivity, year of publication, and reference. This table served as a foundation for various visualization and statistical analyses.

To quantitatively evaluate performance trends and determine the influence of nanoparticle type and food sample matrix on the analytical sensitivity of LFAs, a series of statistical analyses were performed using log_10_-transformed LOD values which were originally reported in various units and converted into ng/mL. Non-parametric Kruskal–Wallis H tests were applied to assess whether significant differences existed in LOD values across various nanoparticle platforms and sample matrix categories, for both singular and multiplex detection formats. To explore potential temporal improvements in assay sensitivity, Spearman’s rank correlation was employed to evaluate the relationship between publication years and reported LOD values. In addition, linear regression analysis of the log-transformed LODs was conducted to estimate the proportion of variance explained by publication year, and R^2^ values reported.

To support interpretation of the extracted data and identify performance trends, a variety of graphical techniques were employed. Donut charts illustrated the proportional distribution of nanoparticle platforms and targeted mycotoxins across the included studies. Scatter plots were generated to visualize the temporal progression of LOD values, stratified by both nanoparticle type and target analyte, highlighting changes in sensitivity and innovation over time. Box plots were used to compare LOD distributions across different food matrices and mycotoxins, enabling visual assessment of central tendency, variability, and outliers in assay performance. All visualizations were systematically categorized based on assay type either singular or multiplex and presented in an organized format to ensure clarity and consistency in data interpretation.

## 3. Results

The review results of nanoparticle types and targeted mycotoxins in LFA studies is illustrated in Panels A and B of Figure 5, respectively.

Panel A depicts the proportionate usage of various nanoparticle types in 78 reviewed LFA studies. AuNPs were the most extensively employed, featuring in 29 studies (37.2%). Fluorescent nanoparticles, including QDs and europium-based nanospheres, were the second most utilized category, each appearing in 15 studies (19.2%). A diverse range of novel or unclassified nanoparticle systems was reported in 12 studies (15.4%), reflecting increasing innovation in LFA nanomaterial design. Hybrid nanomaterials combining optical, magnetic, or catalytic properties were observed in 7 studies (9%), while magnetic nanoparticles (e.g., Fe_3_O_4_) were included in 5 studies (6.4%) to aid in sample pre-concentration. Carbon-based nanomaterials and nanozymes were each reported in 3 studies (3.8%), supporting enhanced signal generation. Metal–organic frameworks (MOFs) and silver nanoparticles (AgNPs) were the least common, each featured in 2 studies (2.6%).

Panel B presents the percentage distribution of mycotoxin groups targeted by LFA platforms across 78 studies. AFs were the most frequently targeted, accounting for 30.1%, followed by ZEN at 21.9%. OTA was represented in 15.8% of cases, and DON in 13.8%. FBs contributed 9.7%, while trichothecenes, including T-2 and HT-2 toxins, were the least frequently assessed, appearing in 8.7% of the reviewed studies.

A dual-panel visualization of the log-scaled LOD values for nanoparticle-based LFAs is presented in Figure 6, summarizing developments in singular mycotoxin detection from 2015 to 2024. Reported LOD values span a broad dynamic range from 10^−5^ ng/mL to 10^3^ ng/mL, indicating significant heterogeneity in analytical sensitivity. Ultra-sensitive detection thresholds (≤10^−5^ ng/mL) appear only sporadically and do not follow a consistent trend over time. There is no observable decrease in LOD values over time, indicating that, despite increasing interest in nanoparticle-based singular detection, assay sensitivity has not systematically improved for any specific mycotoxin. The top panel categorizes log-scaled LOD by target mycotoxin, revealing that AFB1, ZEN, and OTA were the most frequently investigated analytes in singular detection formats. In the bottom panel, which stratifies LODs by nanoparticle type, gold-based nanoparticles remain the most commonly employed platform; however, their performance ranges from femtogram-level sensitivity to values exceeding 1000 ng/mL. This wide distribution underscores that detection sensitivity in singular systems is driven more by assay design parameters, such as antibody quality, surface conjugation, and signal amplification than by the nanoparticle core material itself. In contrast, fluorescent nanoparticles, nanozymes, and carbon-based composites tend to cluster within the moderate to ultra-sensitive range, suggesting potentially better reproducibility in singular formats. Novel nanomaterials such as MOFs, magnetic composites, and hybrid structures have emerged more frequently in recent years, though their detection performance remains highly variable. The most sensitive singular detection system reported achieved a LOD of 0.053 fg/mL (5.3 × 10^−8^ ng/mL) for DON using gold nanoparticles, highlighting the technological potential of well-optimized nanoparticle systems, although such breakthroughs remain isolated.

This exceptional sensitivity was made possible through the construction of a multicomponent SERS tag, comprising AuNR@Ag@SiO_2_–AuNP core–shell–satellite nanoassemblies. The design created ultra-dense electromagnetic “hot spots” via plasmonic coupling between silver-coated gold nanorods and surrounding gold nanoparticles, significantly enhancing the Raman signal. A silica interlayer enhanced colloidal stability and minimized aggregation, which was especially beneficial for maintaining signal integrity in grain matrices. Unlike conventional colorimetric or fluorescence-based methods, the platform employed SERS detection, employing sharp and specific spectral peaks to suppress background noise. The authors also optimized key assay components, including the antibody–nanoparticle conjugation and the LFA architecture, using DON–OVA immobilization at the test line to support competitive binding. Collectively, these innovations established a highly stable and ultra-sensitive LFA platform, representing a major technological leap in mycotoxin biosensing. However, its practical adoption will depend on future validation studies and considerations around device complexity, instrument accessibility and scalability [157].

To quantitatively assess sensitivity trends across singular detection systems, LOD values were categorized into three performance tiers (Figure 7): ultra-sensitive (<0.001 ng/mL), moderate (0.001–1 ng/mL), and low (>1 ng/mL). Only 6.0% of systems achieved ultra-sensitive detection, while 70.0% demonstrated moderate sensitivity and 24.0% fell into the low-sensitivity range. To further examine whether nanoparticle type significantly influenced singular detection performance, a non-parametric Kruskal–Wallis H test was performed, revealing no statistically significant differences in LOD across nanoparticle types (H = 16.26, *p* = 0.1316). This result is consistent with visual observations and supports the conclusion that nanoparticle composition alone does not determine assay performance. Additionally, Spearman’s rank correlation analysis between publication year and LOD produced a weak, non-significant negative correlation (ρ = −0.162, *p* = 0.261), and linear regression of log-transformed LOD values yielded an R^2^ of 0.001 (*p* = 0.824), indicating no meaningful improvement in detection sensitivity over time. Collectively, the visual and statistical analyses of singular detection systems demonstrate that although advances in nanomaterial engineering have expanded the LFA platform, high-performance, ultra-sensitive detection remains highly dependent on system-level factors such as biorecognition elements, surface chemistry, and transduction strategy, rather than nanoparticle type or recency of publication.

To assess trends in analytical sensitivity of multiplex nanoparticle-based LFAs for mycotoxin detection over time, between 2015 and 2024, and across nanomaterial platforms, Figure 8 provides a two-panel visualization of the LOD values (log-transformed, ng/mL) published.

Panel A displays the distribution of LODs by individual target mycotoxins. AFB1 and ZEN were the most frequently detected targets, their LOD values spanning several orders of magnitude. Across all years, most LOD values ranged between 10^−2^ and 10^2^ ng/mL, but a noticeable convergence toward lower values is observed in the most recent publications (2022–2024), suggesting enhanced assay sensitivity. However, despite progress, significant variability persists among targets and platforms, likely reflecting differences in antibody affinity, matrix effects, and assay optimization.

Panel B organizes the same LOD values by nanoparticle type, enabling comparison of material-based performance. AuNPs have remained the most prevalent choice across the decade, due to their reliable optical properties and standardized synthesis protocols. However, the performance of AuNP-based systems was highly variable, LODs ranging from <10^−3^ ng/mL to >10^2^ ng/mL. This heterogeneity suggests that, while AuNPs offer baseline functionality for multiplexing, factors such as antibody affinity, particle-antibody conjugation efficiency, and readout strategy significantly influence overall sensitivity.

In contrast, hybrid nanostructures and nanozymes emerging after 2020 demonstrated consistently low LODs (<0.1 ng/mL) across multiple studies. These multifunctional platforms appear especially suited to multiplex environments, in which the complexity of detecting several targets simultaneously can lead to signal overlap and competition. Their structural and functional adaptability enables enhanced signal discrimination and reduced background interference. Similarly, carbon-based composites and fluorescent nanomaterials, although employed in fewer studies, showed ultra-low LODs in selected assays, highlighting their untapped potential in future multiplex LFIA designs.

A sharp increase in the use of composite and multifunctional materials was noted after 2021, coinciding with significant improvements in detection performance. These platforms are likely benefiting from enhanced surface modification capabilities, better signal-to-noise ratios, and integrated catalytic or photonic properties that amplify detection signals in multiplex conditions.

To provide a quantitative classification of detection sensitivity in these multiplex systems, LOD values were categorized into three performance tiers: ultra-sensitive (<0.001 ng/mL), moderate sensitivity (0.001–1 ng/mL), and low sensitivity (>1 ng/mL) (Figure 9). The majority of systems (68.2%) fell into the moderate sensitivity range, 2.4% achieving ultra-sensitive performance and 29.4% remaining above 1 ng/mL. These findings confirm that, while multiplexing introduces analytical challenges, recent developments are pushing systems toward higher sensitivity with growing consistency.

Statistical analysis supported these trends. A Spearman correlation analysis revealed a significant negative relationship between year of publication and LOD (ρ = −0.357, *p* = 0.0008), indicating temporal improvement in detection performance. Linear regression of log-transformed LODs against publication year yielded an R^2^ of 0.191 (*p* = 2.82 × 10^−5^), suggesting that nearly 19.1% of the variation in LOD values can be attributed to temporal technological advancements. Furthermore, a Kruskal–Wallis H test confirmed a significant influence of nanoparticle type on assay sensitivity (H = 16.95, *p* = 0.031).

Taken together, these findings demonstrate that, while gold nanoparticles remain the most widely used platform in multiplex LFIAs, newer nanomaterials, particularly those with hybrid or multifunctional properties, offer superior sensitivity and adaptability in complex detection environments. Their integration into assay systems, alongside innovations in biorecognition, surface modification, and signal amplification, is contributing to a statistically meaningful enhancement in diagnostic performance across the field. As multiplex LFA technology continues to mature, further progress in nanoparticle engineering and transduction strategies is expected to drive the development of highly sensitive, high-throughput platforms for point-of-care mycotoxin screening.

Figure 10 presents the distribution of LOD values of singular mycotoxin detection assays obtained for
AFB1: The LOD values exhibit substantial variability, ranging approximately from 10^−4^ to nearly 1 ng/mL. The median value for AFB1 is situated around 10^−2^ ng/mL, with the presence of several outliers positioned toward higher concentrations around 10^0^ ng/mL, reflecting significant variability among different assay conditions or analytical platforms.Total Aflatoxin: Observed LOD values range narrowly between approximately 10^−3^ and slightly above 10^−1^ ng/mL, with a median concentration closely centered at about 10^−2^ ng/mL. The tight interquartile range indicates relatively consistent sensitivity across multiple detection methods.Fumonisins: This group exhibits the broadest range, with LOD values extending from slightly above 10^0^ ng/mL up to nearly 10^3^ ng/mL, indicating significantly lower sensitivity compared to other mycotoxins. The median is notably elevated, positioned around 10^2^ ng/mL, emphasizing substantial analytical difficulty in achieving low detection limits.FB1: The LOD values are concentrated within a relatively narrow range, approximately between 10^1^ and 10^2^ ng/mL. The median is closely clustered at around 50 ng/mL, underscoring moderate sensitivity yet less variability compared to the general fumonisin group.ZEN: LOD values span approximately from below 10^−2^ ng/mL to above 10^0^ ng/mL. The median LOD is situated slightly below 10^−1^ ng/mL. Notable outliers are observed, indicating variations possibly influenced by matrix effects or assay-specific conditions.T-2 Toxin: The dataset for T-2 toxin demonstrates tight clustering, ranging narrowly around 10^−1^ ng/mL, with limited variability and few outliers, suggesting relatively uniform assay performance across different analytical setups.DON: The observed range of LOD values for DON is quite wide, spanning from around 10^−1^ ng/mL to above 10^1^ ng/mL. The median lies near 10^0^ ng/mL, with noticeable outliers at higher values, indicating variability likely attributed to assay sensitivity differences and matrix interferences.OTA: LOD values span from below 10^−2^ ng/mL up to slightly above 10^0^ ng/mL, with a median concentration close to 10^−1^ ng/mL. Several outliers towards the higher end suggest variable detection efficiency among reported methods.

Figure 11 presents the distribution of LOD values for singular mycotoxin detection assays across different sample matrices, including cereals and grains, foods and feed, dairy products, and beverages and juices.

Cereals and Grains: The LOD values for cereals and grains demonstrate substantial variability, spanning from approximately 10^−4^ ng/mL to above 10^3^ ng/mL. The median is located near 10^−1^ ng/mL, with numerous outliers observed at higher concentrations (above 10^1^ ng/mL). These outliers suggest notable variability due to potential differences in extraction methods, assay performance, and complex grain matrices.Foods and Feed: The LOD values in foods and feed range from roughly 10^−3^ ng/mL up to just above 1 ng/mL. The median value is close to 10^−1^ ng/mL, indicative of relatively consistent detection performance across these matrices. Few outliers at the higher concentration end highlight moderate variability, likely influenced by assay or matrix-specific differences.Dairy Products: LOD values for dairy products extend from approximately 10^−3^ ng/mL to around 1 ng/mL, with the median concentration positioned slightly below 10^−1^ ng/mL. The presence of isolated outliers at higher values suggests moderate assay variability, potentially influenced by dairy-specific interferences or methodological variations.Beverages and Juices: This category demonstrates the narrowest range, with LOD values closely clustered around 1 ng/mL. The median is nearly identical to this value, suggesting minimal variability and high consistency in analytical sensitivity across beverage and juice matrices.

A statistical analysis was performed to assess whether the type of sample matrix significantly influences the analytical sensitivity, measured as the LOD, of singular detection LFAs for mycotoxins. Due to the positively skewed and non-normal distribution of the raw LOD data, all statistical analyses were conducted using log_10_-transformed values (log_10_ LOD), consistent with standard reporting practices in LFA evaluation.

A Kruskal–Wallis H-test was conducted to compare the distributions of LOD values across four distinct sample matrices: cereals and grains, foods and feed, dairy products, and beverages and juices. The analysis revealed no statistically significant differences among these matrices (H = 5.04; *p* = 0.169), suggesting that the composition of the sample matrix does not meaningfully affect the analytical sensitivity of singular detection LFAs within this dataset.

Figure 12 presents the distribution of LOD values for seven mycotoxins AFB1, FB1, DON, OTA ZEN, AFM1, and T-2 toxin as determined by multiplex LFAs. The data are plotted on a logarithmic scale (ng/mL) to capture the broad dynamic range observed across different analytes.

AFB1 demonstrated excellent assay sensitivity, with LOD values ranging from approximately 10^−4^ to 10^1^ ng/mL, and a median centered around 10^−1^ ng/mL. The narrow IQR and the consistent clustering of values indicate robust and reproducible detection performance for this high-priority mycotoxin.In the case of FB1, LOD values spanned a broader range, from approximately 10^−2^ to over 10^2^ ng/mL, with a median value near 10^0^ ng/mL. The presence of several high-value outliers suggests increased variability in assay performance, potentially due to matrix effects or structural differences in the analyte affecting antibody binding efficiency.DON exhibited the widest variability in LODs among all the analytes, with values extending from 10^−1^ to nearly 10^3^ ng/mL, and a median around 10^1^ ng/mL. This substantial spread and elevated central tendency highlight challenges in achieving consistent sensitivity for DON using multiplex platforms, likely attributable to its hydrophilic nature and weaker immunogenic profile.OTA showed a highly favorable detection profile, with LODs ranging from 10^−4^ to 10^0^ ng/mL and a median near 10^−1^ ng/mL. The narrow IQR and absence of high outliers reflect high reproducibility and minimal interference across matrices, underscoring the assay’s capacity for reliable OTA detection.ZEN demonstrated a moderate spread in LODs, ranging from 10^−2^ to 10^1^ ng/mL, with a median near 10^−1^ ng/mL. While a few outliers were observed, the majority of values clustered within a consistent range, indicating satisfactory sensitivity and reliability.AFM1, a critical biomarker for dairy safety, exhibited the lowest LOD values, ranging from 10^−3^ to just below 10^0^ ng/mL, and a median around 10^−2^ ng/mL. The compact distribution and lack of extreme outliers confirm high assay sensitivity and reproducibility for this analyte in milk-based matrices.T-2 toxin displayed a moderate range of LOD values between 10^−2^ and 10^2^ ng/mL, with the median around 10^0^ ng/mL. While the interquartile range suggests acceptable consistency, the presence of broader values indicates potential assay limitations in certain sample types or concentrations.

Overall, the multiplex LFAs demonstrated excellent sensitivity and precision for AFB1, OTA, and AFM1, while FB1, ZEN, and T-2 showed moderate variability. DON posed the greatest challenge in detection sensitivity, underscoring the need for further optimization in multiplex assay configurations for this mycotoxin. These findings validate the capability of multiplex LFA systems to deliver rapid, sensitive, and multi-analyte detection suitable for diverse food safety monitoring applications.

Figure 13 presents the distribution of LOD values for mycotoxins in three different food matrices cereals and grains, foods and feed, and dairy as determined by multiplex LFAs. The LODs are expressed on a logarithmic scale (ng/mL) to account for the wide dynamic range of detection sensitivities across matrices.

Cereal and grain matrices exhibited the widest range of LOD values, with values spanning from 10^−4^ ng/mL to over 10^1^ ng/mL, and a median near 10^−1^ ng/mL. A large number of high-value outliers were observed, some extending close to 10^3^ ng/mL, indicating substantial variability in assay sensitivity.Foods and feed samples showed a narrower LOD distribution, ranging from approximately 10^−3^ to 10^0^ ng/mL, with a median around 5 × 10^−2^ ng/mL. Although a few outliers were observed, the tighter interquartile range suggests relatively consistent assay performance across different food and feed types.Dairy matrices demonstrated the highest sensitivity and reproducibility, with LODs tightly clustered between 10^−2^ and 10^−1^ ng/mL, and a median around 3 × 10^−2^ ng/mL. No outliers were detected, and the narrow IQR reflects uniform detection performance. This may be attributed to the liquid nature of dairy samples and effective pre-treatment strategies such as centrifugation or protein precipitation, which reduce matrix interference.

A statistical analysis was conducted to determine whether the type of sample matrix significantly influences analytical sensitivity, expressed as the LOD, in multiplex LFAs for mycotoxins. Due to the positively skewed and non-normal distribution of the raw LOD data, all values were log_10_-transformed prior to analysis. A Kruskal–Wallis H-test was applied to compare the distributions of log-transformed LOD values across three distinct sample matrices: cereals and grains, foods and feed, and dairy products. The test yielded a statistic of H = 1.20 with a corresponding *p*-value of 0.549, indicating that there were no statistically significant differences in analytical sensitivity among the different matrices. This suggests that the type of sample matrix does not have a meaningful effect on the sensitivity of multiplex LFAs within the context of this dataset.

## 4. Discussion

As illustrated in Figure 5A, AuNPs constitute the most widely utilized nanomaterial in LFA platforms for mycotoxin detection, representing 37.2% of the studies reviewed. This dominance is closely linked to their exceptional optical characteristics, especially their localized surface plasmon resonance (LSPR), which produces distinct colorimetric signals visible to the naked eye [201]. This inherent optical responsiveness enables rapid and low-cost detection of target analytes without the need for complex instrumentation, making AuNP-based LFAs particularly suitable for point-of-care and field applications [122,123,201]. The high surface-area-to-volume ratio of AuNPs provides abundant sites for functionalization, allowing for efficient conjugation with antibodies [134], aptamers [125,148], or other biorecognition molecules. This not only enhances the specificity of detection but also significantly lowers the LOD across a broad range of mycotoxins [201,202]. Their chemical stability, biocompatibility, and ease of synthesis further support their widespread adoption in commercial and research settings [201,202]. Recent trends show increasing integration of AuNP-based LFAs with external optical readers and smartphone-based imaging systems, improving the quantitative accuracy of colorimetric readouts [199,201,202]. These advancements are driven by both the decreasing cost of portable readers and the growing demand for digital, user-friendly diagnostics, especially in low-resource environments. Further innovation involves the use of core–shell architectures and hybrid nanostructures, where AuNPs are combined with materials like quantum dots [183] or magnetic nanoparticles [144]. These configurations facilitate multiplexed detection and signal enhancement, expanding the analytical capabilities of LFA platforms [203,204,205,206]. Moreover, their compatibility with emerging smartphone-integrated diagnostic tools aligns with the future direction of personalized, real-time, and decentralized testing [203,204,205,206].

Fluorescent nanoparticles (FNPs), including QDs and europium-based nanospheres, emerged as the second most utilized nanomaterial type in LFA studies, accounting for 19.2% of reviewed cases. Their increased adoption is largely driven by their superior optical properties such as high quantum yield, photostability, and tunable emission wavelengths that enable both enhanced sensitivity and multiplexed detection within a single test strip [131,140,149,154,179,192]. These features are particularly advantageous in detecting co-contaminants and achieving lower LOD across a wide range of food matrices. Among these, europium-based nanospheres have gained prominence in time-resolved fluorescence (TRF) assays, in which measurement of delayed fluorescence emission minimizes background interference and improves the signal-to-noise ratio. Applications of TRF using Eu(III) chelate-doped beads [179] and Eu/Tb(III) hybrid systems [192] have demonstrated ultralow detection limits across complex food matrices. Recent smartphone-enabled formats employing EuNPs for the detection of OTA and FB1 further underscore their suitability for portable field diagnostics [140,168]. In parallel, QDs provide sharp, size-dependent emission spectra with negligible spectral overlap, which is ideal for multi-target quantification in fluorescence-based multiplex LFAs [131,149,183,196,197]. Studies using CdSe/ZnS [137], CdSe/CdS/ZnS [196], and cadmium-free InP/ZnS [197] QDs confirm their broad compatibility with food and feed matrices, despite ongoing concerns around cytotoxicity and environmental safety for cadmium-based materials. UCNPs, such as Lu^3+^- or NaYF_4_:Yb,Er-doped particles, offer deep-tissue penetration and anti-Stokes emission advantages, enhancing performance in visually or spectrally complex samples [146,154,164]. Fluorescent microspheres, as exemplified in red-emissive formats [127,151], also contribute to improved assay brightness, lower LODs, and compatibility with both qualitative and quantitative readouts. Additionally, composite materials such as CdSe/SiO_2_ QBs [189] and phycocyanin-labeled latex nanospheres [178] have facilitated simultaneous multi-analyte detection with reduced cross-reactivity.

Despite the need for specialized readers and higher production costs, the superior analytical performance of FNPs consistently surpasses that of conventional colorimetric systems. Collectively, these advances position fluorescent nanoparticles as a versatile and powerful class of nanolabels, central to the ongoing evolution of highly sensitive, multiplexed, and portable LFA platforms. Approximately 15.4% of reviewed LFA studies were categorized under novel or unclassified nanoparticle systems, highlighting a growing interest in advanced nanostructures designed to overcome limitations of traditional materials (Figure 5A). This group includes gold nanobipyramids [161], which exhibit efficient photothermal conversion and enable temperature-responsive signal readouts through thermal modulation mechanisms, thereby improving sensitivity and enabling alternative detection strategies in optically challenging matrices. Core–shell–satellite nanoassemblies [157] offer highly active SERS platforms with enhanced structural stability, enabling ultra-trace detection with improved signal-to-noise ratios. Inner-filter effect (IFE) platforms [175], which exploit spectral overlap between donor and absorber components, modulate fluorescence intensity to reduce background interference and enable precise signal control. Photothermally active composites such as BP–Au [143] and Cu_2−_xSe–Au [187] function by converting light into localized heat, thereby facilitating thermometric signal modulation and amplifying detection signals through temperature-dependent changes in optical properties. Collectively, these systems enhance detection sensitivity, signal stability, and multiplexing capabilities. A key advantage lies in their ability to address persistent challenges such as turbid matrix interference, multi-analyte discrimination, and dual-mode detection. By shifting away from traditional reliance on direct colorimetric or fluorescent signals which are often distorted by scattering and absorption in opaque or particulate-rich samples, these platforms utilize thermally modulated, spectrally distinct, or fluorescence-controlled mechanisms that are inherently more robust in complex matrices. As a result, they offer reliable performance in real-world sample conditions and contribute to the development of next-generation, field-deployable diagnostic tools.

Additionally, a smaller yet scientifically meaningful subset of LFA studies representing approximately 19.2% of the reviewed dataset has focused on the use of specialized nanomaterial systems beyond conventional AuNPs (Figure 5A). These include magnetic nanoparticles (MNPs, ~6.4%), carbon-based nanomaterials (~3.8%), nanozymes (~3.8%), metal–organic frameworks (MOFs, ~2.6%), and AgNPs (~2.6%). These nanomaterials are not only less common but are strategically selected to enhance application-specific performance, particularly in challenging food matrices requiring improved sensitivity, multiplexing, or field-deployability. Their engineered physicochemical properties such as magnetic enrichment, catalytic amplification, or optical tunability allow them to overcome key limitations in traditional LFA systems. The following subsections explore each material category in detail, highlighting their distinct contributions to improved assay performance in food-based mycotoxin detection.

MNPs such as Fe_3_O_4_ and carboxyl-coated magnetic silica were utilized in approximately 6.4% of studies and serve multiple roles including sample pre-concentration, purification, and automated movement in microfluidic contexts. For instance, carboxylated Fe_3_O_4_ MNPs were employed in a dual-mode LFIA for the detection of ZEN, enabling visual and smartphone-based quantification through magnetic enrichment and enhanced signal recovery [144]. Similarly, aptamer-modified MNPs were used in a “three-in-one” platform that facilitated enrichment, purification, and detection of OTA in a single test, minimizing handling steps and improving assay robustness in turbid beverage samples [162]. Emerging magnetic systems also integrate multichannel detection capabilities, such as platforms utilizing Fe_3_O_4_@PEI/AuMBA@Ag-MBA nanocomposites, which combine magnetic enrichment with bi-channel SERS-based detection of both AFB1 and ZEN [181]. These multifunctional MNP-based systems offer considerable potential for automated, portable, and multiplexed LFA applications.

Carbon-based nanomaterials, although accounting for only 3.8% of the studies reviewed, have gained increasing attention in LFA development due to their unique combination of optical tunability, high electronic conductivity, and exceptional chemical stability. These materials, including amorphous carbon nanoparticles and carbon-containing metal–organic framework-derived hybrids, have been successfully applied across multiple assay formats. Their versatility supports various detection mechanisms, such as colorimetric, fluorescent, and photothermal readouts, making them adaptable to both single-analyte and multiplexed detection contexts [124,188]. In addition to their favorable signal-to-noise ratios, carbon-based nanomaterials are often valued for their cost-effectiveness, thermal resilience, and chemical inertness, attributes that make them especially suitable for field-deployable diagnostics in resource-limited or high-temperature environments. As the demand for high-performance and scalable biosensing platforms increases, these materials are expected to play a growing role in next-generation LFA technologies.

Nanozyme-based nanoparticles, which mimic the catalytic activity of natural enzymes, have emerged as potent alternatives for signal amplification in lateral flow assays, representing approximately 3.8% of the studies reviewed. These materials offer several advantages over biological enzymes, including enhanced thermal and chemical stability, extended shelf life, and lower production costs, making them highly suitable for robust, point-of-care diagnostic platforms. For example, noble metal-based nanozymes such as Pt@Au nanoflowers, combined with horseradish peroxidase (HRP), have enabled dual-enzyme amplification strategies that significantly enhance the sensitivity of ZEN detection [142]. Similarly, CuCo@PDA nanozymes, exhibiting peroxidase-like activity, have been integrated into dual-mode lateral flow formats, supporting both visual and smartphone-based quantification for AFB1 detection [126]. More recently, Fe–N–C single-atom nanozymes (SAzymes) have been applied for the multiplexed detection of AFB1 and FB1, offering high catalytic activity and strong target affinity in enzyme-free systems [172]. These developments highlight the growing potential of nanozymes as high-performance alternatives in next-generation LFA technologies.

Metal–organic frameworks (MOFs) are highly porous, crystalline materials characterized by their exceptional surface area, tunable pore structure, and chemical functionality, making them attractive candidates for signal amplification in biosensing. Although they appeared in only 2.6% of the reviewed studies, MOFs have demonstrated considerable potential in LFAs due to their structural versatility and ability to form multifunctional hybrid platforms [132,133,207,208]. In LFA applications, MOFs have been employed primarily as signal enhancers, where their catalytic activity or loading capacity can amplify both colorimetric and fluorescent signals. For instance, polydopamine-coated MOF composites have been used to enhance the visual response in immunoassays, employing the high surface area and coordination sites of MOFs for efficient probe immobilization and catalysis [132]. Other hybrid formats have combined MOFs with aggregation-induced emission (AIE) luminogens to enable dual-modal detection, offering simultaneous visual and fluorescence outputs with improved sensitivity in complex food matrices [133]. These approaches highlight the adaptability of MOFs in developing robust, multiplexed, and sensitive LFA platforms, positioning them as promising materials for next-generation point-of-care diagnostics.

AgNPs are known for their superior surface plasmon resonance (SPR) characteristics, offering sharper absorption peaks and stronger signal amplification than AuNPs. However, their use in LFAs remains limited, accounting for only 2.6% of reviewed studies, largely due to challenges related to oxidation, cytotoxicity, and colloidal instability, particularly under variable environmental conditions where LFAs are typically applied [165]. To overcome these limitations, researchers have explored core–shell nanostructures such as Ag@Au nanoparticles, which synergize the optical advantages of both metals for ultrasensitive and multiplexed SERS-based detection of toxins like AFB1 and OTA [171]. Additionally, fluorescence-quenching LFIAs incorporating AgNPs have demonstrated improved sensitivity for OTA detection in complex matrices such as wine and juice [165]. In hybrid approaches, AgNPs have also been combined with quantum dots and AuNPs to enhance signal resolution and detection accuracy in assays for mycotoxins like fumonisin [137].

These studies highlight the promising analytical performance of AgNPs, particularly when embedded in composite or stabilized platforms. Their broader adoption in LFA diagnostics will depend on the continued development of protective coatings and biocompatible modifications to improve their environmental robustness and biosafety.

The donut chart presented in Figure 5B offers valuable insight into the distribution of mycotoxins targeted by 78 reviewed LFA studies, revealing clear patterns in both research intensity and diagnostic prioritization. AFs emerged as the most frequently targeted mycotoxins, identified in 30.1% of the studies. This strong representation likely reflects the widespread contamination of maize, nuts, spices, and cereals by AFB1, a potent hepatotoxic Group 1 carcinogen [2,209,210]. AFB1 disrupts protein synthesis, induces oxidative stress, and causes DNA damage, and its persistent presence in food products highlights the critical need for rapid screening methods like LFAs. Aflatoxins are also subject to strict international regulatory limits, reinforcing their prioritization in assay development [2,27,209,210].

ZEN was targeted in 21.9% of the studies. ZEN is well known for its estrogenic activity, leading to reproductive and hormonal disorders, especially in livestock and humans [26,27]. Its frequent inclusion in LFA studies may be attributed to both biological relevance and the feasibility of generating specific antibody responses, making it suitable for integration into multiplex diagnostic formats.

OTA was represented in 15.8% of LFA studies. A potent nephrotoxin, OTA is linked to renal failure, and renal cancers [211]. Found in dried fruits, cereals, coffee, and wine, OTA remains a serious food safety concern due to its chronic toxicity and low permissible thresholds. Its representation in LFA systems suggests increased interest in on-site detection for consumer safety and export assurance [211].

DON appeared in 13.8% of studies. This type B trichothecene targets the gastrointestinal and immune systems, particularly affecting rapidly dividing cells [212]. Clinical effects include vomiting, diarrhea, anorexia, and immune suppression. Although DON is more prevalent in temperate climates, its consistent inclusion in LFA studies reflects its high incidence in cereal crops such as wheat and barley [212]. FBs were detected in 9.7% of studies. FB1 is associated with severe toxic effects, including neural tube defects, hepatotoxicity, and stunted growth in children [213,214]. FB1’s association with esophageal cancer and contamination in maize-based diets, particularly in developing regions, underlines the importance of accessible point-of-care diagnostics, although its lower LFA representation may reflect technical assay limitations or under-monitoring in some regions [213,214,215].

T-2 and HT-2 toxins, representing type A trichothecenes, were the least detected, included in only 8.7% of LFA studies. These mycotoxins are highly cytotoxic, targeting hematopoietic tissues and causing immunosuppression, hemorrhaging, and gastrointestinal distress [216]. Their limited detection may reflect antibody development challenges, lower routine monitoring in food safety programs, or lower reported incidence.

Altogether, this distribution reflects the intersection of epidemiological risk, toxicological severity, and regulatory enforcement. The dominance of AFs, ZEN, and OTA suggests a concentrated research effort toward those mycotoxins most consistently associated with health outcomes and trade compliance. In contrast, the underrepresentation of T-2/HT-2 indicates a need to expand analytical coverage in LFA design to capture emerging or co-occurring threats especially in stored grains where synergistic toxicity with other mycotoxins has been observed.

The current systematic review provides a comparative analysis of the evolution of singular and multiplex nanoparticle-based LFAs for mycotoxin detection over the period 2015–2024. While both formats have seen increased use of advanced nanomaterials such as AuNPs, nanozymes, and hybrid composites, their developmental paths diverge significantly. Multiplex LFAs have demonstrated a statistically significant improvement in analytical sensitivity over time (Figure 8), recent platforms consistently reporting lower LODs and greater reproducibility. In contrast, singular LFAs (Figure 6) display persistent variability in LODs across studies and show no systematic trend toward enhanced sensitivity, despite the integration of novel nanomaterials in many recent designs.

As illustrated in Figure 6, singular LFAs demonstrate a wide range of detection limits across different studies, with no clear downward trend in LOD values over time. This observation is supported by statistical analysis, which shows a weak, non-significant correlation between publication year and sensitivity. Moreover, the type of nanoparticle used did not significantly affect detection performance, suggesting that core material properties are not the primary drivers of analytical sensitivity in these systems.

Instead, assay performance appears to depend heavily on system-level factors such as antibody affinity, surface conjugation chemistry, membrane flow properties, and signal amplification strategies. Even widely used AuNPs displayed a broad spectrum of performance outcomes, highlighting the importance of optimized assay integration.

Notably, a small number of singular platforms have achieved exceptional sensitivity when material and assay design were carefully co-optimized. One such example is a SERS-based platform employing AuNR@Ag@SiO_2_–AuNP nanoassemblies, which achieved femtogram-level LODs for DON [157]. This platform combined dense electromagnetic hotspot formation with silica-mediated colloidal stability and optimized antibody–nanoparticle conjugation, representing a significant technological advancement. However, these breakthroughs remain technically complex and are not yet widely scalable.

Other emerging materials, including fluorescent nanoparticles [131,154,158], nanozymes [126,153], and magnetic composites [144,162], have shown potential for improving sensitivity in singular LFAs. However, their performance varies considerably, as shown in Figure 6B, largely due to inconsistencies in surface functionalization, matrix effects, and limited assay standardization.

As indicated in Figure 7, only a small fraction of singular assays reached ultra-sensitive detection levels (<0.001 ng/mL), with the majority demonstrating moderate sensitivity. Future improvements will require a shift from focusing solely on new nanomaterials to optimizing bioconjugation techniques, test-line architecture, and signal transduction. Incorporating smartphone-based readers and AI-enabled signal analysis could also enhance quantification and reproducibility in decentralized testing environments.

In contrast to singular systems, multiplex LFAs have demonstrated measurable gains in analytical performance over time. As shown in Figure 8A, recent studies increasingly report lower LOD values, supported by a statistically significant correlation between publication year and improved sensitivity. These improvements are largely driven by the integration of multifunctional nanomaterials that enable more efficient signal discrimination and amplification in complex detection environments.

Figure 8B illustrates how nanoparticle type significantly affects performance in multiplex platforms unlike in singular systems indicating that material properties play a more decisive role when multiple targets are involved. AuNPs remain the most common detection label due to their optical reliability and production scalability. However, their sensitivity varies greatly depending on factors such as antibody loading efficiency and signal readout integration.

By contrast, hybrid nanostructures, nanozymes, and photothermally responsive composites have demonstrated more consistent low LODs (<0.1 ng/mL) across multiple analytes [143,153,157,175,187]. For example, BP–Au and Cu_2−_xSe–Au platforms have utilized temperature-based detection modes to reduce background interference and improve analyte resolution [143,187]. Similarly, IFE-based systems have enabled simultaneous multi-target detection by combining red-emitting quantum dots with engineered gold nanoparticles [175].

As shown in Figure 9, while most multiplex assays currently fall within the moderate sensitivity range (0.001–1 ng/mL), a growing number are achieving low and ultra-sensitive thresholds. This trend coincides with the increased use of composite materials and reflects a maturation of multiplex LFA design. However, challenges such as cross-reactivity, analyte competition, and membrane crowding remain, and achieving reproducible ultra-sensitive detection across multiple targets continues to be technically demanding.

To address these issues, future multiplex platforms should incorporate modular test-line architectures, vertical-flow designs, and improved probe spatial separation. Additionally, the integration of smartphone-based readout systems, AI-assisted signal interpretation tools, and standardized calibration protocols will be essential for ensuring analytical reproducibility, particularly in field-deployable formats.

Figure 10, Figure 11, Figure 12 and Figure 13 provide a detailed assessment of analytical sensitivity in singular and multiplex LFAs for mycotoxin detection across different analytes and food matrices. Singular LFA performance varied substantially across mycotoxins, as illustrated in Figure 10. AFB1 and total aflatoxins consistently achieved low LODs with narrow IQRs, confirming their suitability for rapid on-site detection. This is supported by a previous study [217], which reported that most studies achieved LODs between 0.01 and 1.0 ng/mL for AFB1, emphasizing its reliable detectability in LFA formats.

In contrast, DON and fumonisins displayed the highest LODs and the broadest variability, with FB1 showing moderate sensitivity and less dispersion within the group. These detection challenges likely stem from their hydrophilic nature, weak immunogenicity, and matrix-induced interference.

Figure 11 expands the evaluation of singular LFAs by comparing analytical sensitivity across four sample matrix categories: cereals and grains, foods and feed, dairy products, and beverages and juices. Cereals and grains showed the widest LOD range, from 10^−4^ to above 10^3^ ng/mL, with a median near 10^−1^ ng/mL and multiple high-end outliers. This likely reflects interference from complex matrix components such as starches, proteins, and inconsistent contamination. In contrast, foods and feed, and dairy matrices exhibited more consistent performance, with LODs largely between 10^−3^ and 1 ng/mL. Beverages and juices demonstrated the tightest clustering, likely benefiting from simpler matrix composition and improved analyte diffusion. Despite these visual trends, a Kruskal–Wallis H-test (H = 5.04; *p* = 0.169) revealed no statistically significant differences between matrix types, suggesting that while sample complexity can introduce variability, it does not consistently impact overall assay sensitivity.

Figure 12 presents the distribution of LOD values for mycotoxins in multiplex LFAs. AFB1, OTA, and AFM1 showed the most favorable sensitivity, with low medians and tightly clustered values. AFB1 exhibited LODs down to 10^−4^ ng/mL and AFM1 around 10^−2^ ng/mL, confirming strong assay precision. OTA followed a similar profile with minimal variability. In contrast, DON had the broadest LOD range spanning nearly four orders of magnitude and the highest median near 10^1^ ng/mL, reaffirming its detection difficulty in multiplex contexts. FB1 and T-2 also displayed broader distributions and moderate sensitivity, while ZEN maintained an intermediate profile with occasional outliers. These results reinforce that detection efficiency in multiplex LFAs is primarily analyte-dependent, and certain targets like DON consistently challenge assay robustness due to their chemical properties.

Matrix effects in multiplex LFAs, presented in Figure 13, followed similar trends to those in singular assays. Dairy-based samples yielded the most uniform results, with LODs between 10^−2^ and 10^−1^ ng/mL and no outliers, pointing to effective pre-treatment and homogeneity. Cereals again showed a wide LOD range, while foods and feed represented a middle ground. Statistical analysis (H = 1.20; *p* = 0.549) confirmed no significant differences among matrices, indicating that reproducibility is more closely tied to analyte chemistry and assay design than matrix category.

Taken together, these findings indicate that LFA sensitivity is primarily governed by the physicochemical properties of the target mycotoxins rather than the sample matrix. Persistent challenges with DON and fumonisins highlight the need for improved antibody generation, conjugation techniques, and signal amplification strategies tailored to difficult targets.

From an application perspective, the demonstrated robustness of LFAs, particularly in dairy and liquid matrices, supports their use as practical tools for real-time, multi-analyte food safety monitoring. Future work should prioritize assay standardization, enhanced quantitative readout technologies, and matrix-aware protocol development to ensure reproducibility across heterogeneous sample types.

Moreover, several LFA studies, both singular and multiplex, have demonstrated LOD values that exceed internationally accepted regulatory thresholds, raising concerns about their readiness for practical deployment in food safety monitoring. For instance, a 2021 singular detection study reported an AFB1 LOD of 3.4 μg/kg using red fluorescent microspheres, exceeding the EU’s regulatory limit of 2 μg/kg for groundnuts intended for direct human consumption. In another case, a 2023 multiplex study using AuNP nanobipyramids reported an OTA LOD of 20 ng/mL (20 μg/kg), surpassing the EU threshold of 3 μg/kg for processed cereal products [161]. These discrepancies reflect the ongoing limitations in sensitivity among some fluorescence-based and nanoparticle-enhanced LFA systems when applied to complex food matrices. More notably, in the domain of fumonisin detection, a 2019 multicolor ICST multiplex assay targeting FB1 showed an LOD of 1000 μg/kg [169]. While this value is equivalent to the regulatory maximum for maize-based snacks and breakfast cereals (1000 μg/kg), it may not meet stricter thresholds applied to certain other maize-derived products. Meanwhile, a 2019 study using AuNPs in maize grain reported a fumonisin LOD of 4000 µg/kg [141], which exceeds the maximum permissible levels established for fumonisins in all maize categories, including unprocessed maize (2000 µg/kg), maize flour (1000 µg/kg), and maize-based baby foods (200 µg/kg) [31]. Furthermore, a 2024 aptamer-based three-channel assay targeting AFM1 in milk reported an LOD of 0.39 ng/mL significantly above the Codex and EU-mandated limit of 0.05 μg/kg for milk [174]. Such findings highlight the importance of aligning LFA analytical sensitivity not just with general regulatory ceilings, but also with product-specific thresholds defined in official food categories. As LFA platforms evolve, it remains essential to distinguish exploratory or research-stage methods from those intended for official monitoring or commercial deployment. Only assays rigorously validated against matrix-specific limits should be considered compliant with food safety regulation [30,189,190,191].

## 5. Conclusions

This study provides a systematic and data-driven evaluation of nanoparticle-based LFAs for the detection of mycotoxins across singular and multiplex formats from 2015 to 2024. The findings demonstrate that while AuNPs remain the most widely adopted label due to their robust optical properties and ease of surface functionalization, their analytical sensitivity varies significantly depending on system-level design factors such as antibody affinity, signal amplification strategies, and matrix interference control. Fluorescent nanoparticles, nanozymes, and carbon-based composites, especially in multiplex configurations, have shown promise in improving detection limits and assay reproducibility, particularly for high-priority toxins such as AFB1, ZEN and OTA.

Despite substantial progress in nanomaterial innovation, the analysis reveals that improvements in analytical sensitivity have not been uniformly achieved. Singular LFAs largely operate within a moderate sensitivity range suitable for routine regulatory screening, with only a small subset achieving ultra-trace detection levels. In contrast, multiplex LFAs exhibit more favorable trends in detection performance, particularly in recent years, with statistically significant improvements attributed to methodological advancements and the incorporation of multifunctional nanomaterials.

Importantly, across both assay types, sample matrix composition did not significantly influence LOD performance, as evidenced by non-significant Kruskal–Wallis results. This matrix-independence supports the robustness and versatility of LFA platforms for use across diverse food products, including complex substrates such as cereals and dairy.

However, the growing number of new nanoparticle systems and detection methods highlights an important issue: many current LFAs still do not meet the strict safety limits set by international organizations such as the EU and WHO. Furthermore, variability in commercial manufacturing parameters such as the synthesis of nanomaterials, antibody conjugation efficiency, membrane composition, and strip assembly techniques can markedly influence the analytical performance, sensitivity, and reproducibility of LFAs. These disparities underscore the necessity for standardized production protocols and stringent quality assurance measures to ensure consistent assay reliability across different manufacturing sources. The wide interest in developing new techniques suggests that existing tests, especially those designed for detecting a single mycotoxin, often lack the required sensitivity, reliability, and ability to perform well in real-world food samples. This performance gap is not only a technical issue but also a concern for public health. Closing this gap will require ongoing innovation and a complete review of how these tests are designed, to ensure they consistently meet food safety standards.

In conclusion, future advancements in LFA-based mycotoxin detection should move beyond a material-centric focus and adopt a holistic engineering approach. Priorities must include the development of high affinity biorecognition elements, improved signal transduction mechanisms, and standardized validation protocols to ensure reproducibility and scalability. Key priorities include the development of biorecognition elements with high affinity and specificity such as optimized antibodies, aptamers, or nanobodies to enhance assay selectivity and sensitivity. Equally important is the optimization of membrane flow dynamics and strip architecture to ensure consistent fluid migration and reliable performance.

Further improvements should address assay kinetics and streamline sample preparation, particularly through the integration of on-strip filtration or matrix-cleaning components to improve compatibility with diverse food samples. The incorporation of digital tools, including smartphone-assisted detection, automated image processing, and mobile-based data platforms, can support real-time interpretation and effective field-based data management.

Enhancing usability through provision of clear visual instructions, QR-linked instructions, and simplified result formats will further increase accessibility for a broader range of users. Additionally, the development of portable, stable test kits suitable for varying environmental conditions will extend the applicability of these assays. Finally, implementing standardized validation protocols and harmonized calibration procedures is essential to ensure reproducibility, comparability, and regulatory acceptance across different settings.

Such integrative strategies will be critical for the realization of next-generation LFAs that are not only ultra-sensitive and multiplexed, but also portable, cost-effective, and suitable for deployment in real-world food safety monitoring and public health interventions. Collaborative efforts among researchers, manufacturers, and regulatory bodies will be essential to accelerate the translation of laboratory-scale innovations into commercially viable and globally accepted diagnostic tools.

## Figures and Tables

**Figure 1 toxins-17-00348-f001:**
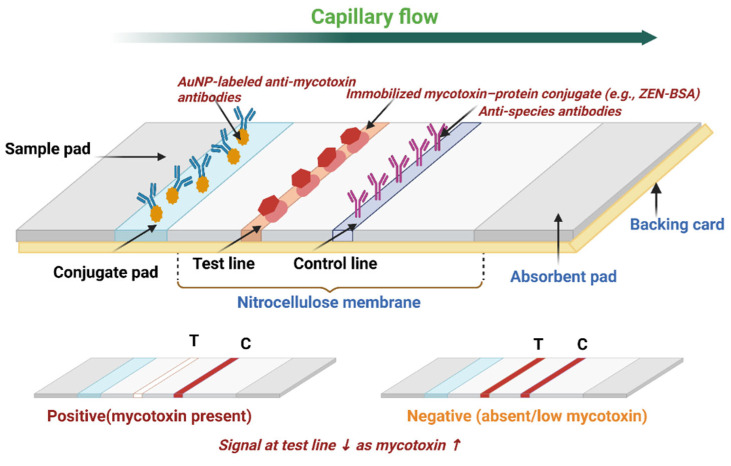
Schematic of a competitive lateral flow assay (LFA) for mycotoxin detection. The assay consists of a sample pad, conjugate pad, nitrocellulose membrane with test (T) and control lines (C), absorbent pad, and backing card. AuNP-labeled anti-mycotoxin antibodies migrate with the sample by capillary flow and interact either with free mycotoxin analytes (hapten) in a solution or with the immobilized mycotoxin–protein conjugate (e.g., ZEN–BSA) at the test line. In the presence of mycotoxin, free analytes compete for antibody binding, resulting in a reduced or absent signal at the test line, while the control line remains visible, confirming assay validity and a positive result. The signal intensity at the test line is inversely proportional to the mycotoxin concentration in the sample (bottom left). In contrast, when the mycotoxin is absent, antibody–nanoparticle conjugates bind to the immobilized conjugate, generating a visible signal at the test line alongside the control line, indicating a negative result (bottom right). Red arrows indicate that signal intensity at the test line decreases as mycotoxin concentration increases (created with Biorender).

**Figure 2 toxins-17-00348-f002:**
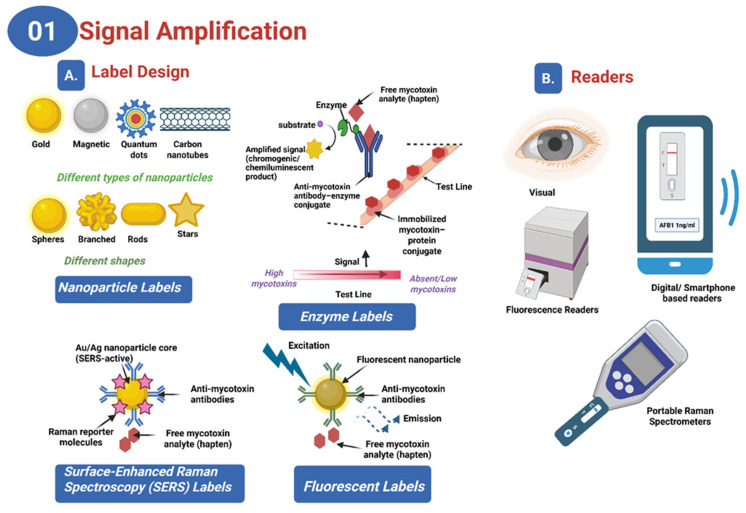
Schematic representation of label designs (**A**) and detection readers (**B**) used for signal amplification in lateral flow assays for mycotoxin detection. Labeling strategies include nanoparticles, enzymes, surface-enhanced Raman spectroscopy (SERS), and fluorescent labels. Reader technologies range from visual inspection to portable optical, and Raman devices, with smartphone-based digital readers offering field-deployable quantification (created with Biorender).

**Figure 3 toxins-17-00348-f003:**
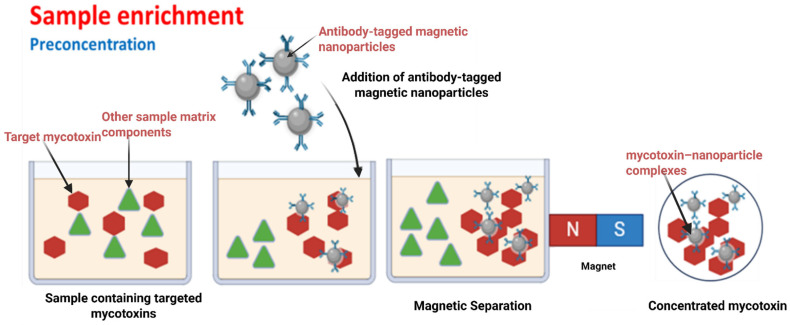
Sample enrichment using magnetic nanoparticles for mycotoxin detection in LFAs.

**Figure 4 toxins-17-00348-f004:**
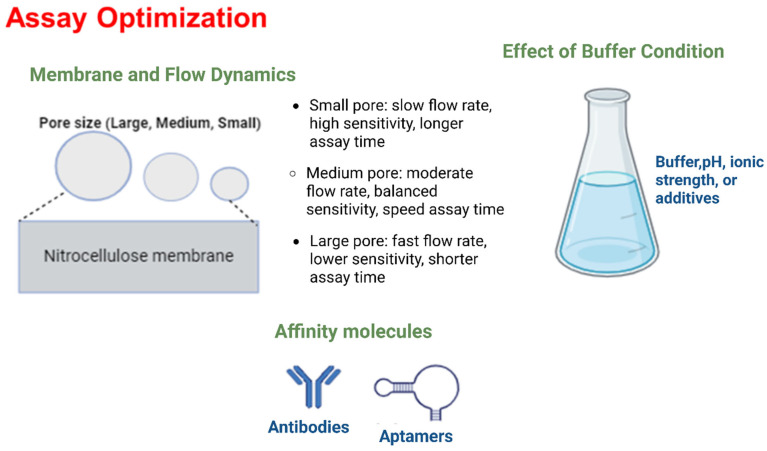
Factors affecting the optimization of lateral flow assays (LFAs) through flow dynamics, membrane pore sizes, affinity molecules, and buffer conditions (created with Biorender).

**Figure 5 toxins-17-00348-f005:**
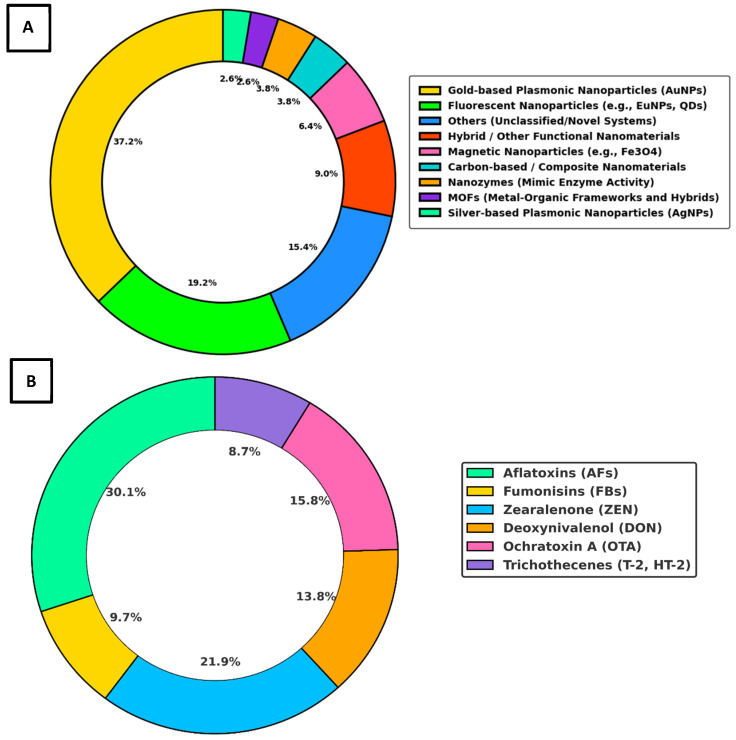
Distribution of studies of different nanoparticle (**A**) types and mycotoxin detection (**B**) in LFA (*n* = 78).

**Figure 6 toxins-17-00348-f006:**
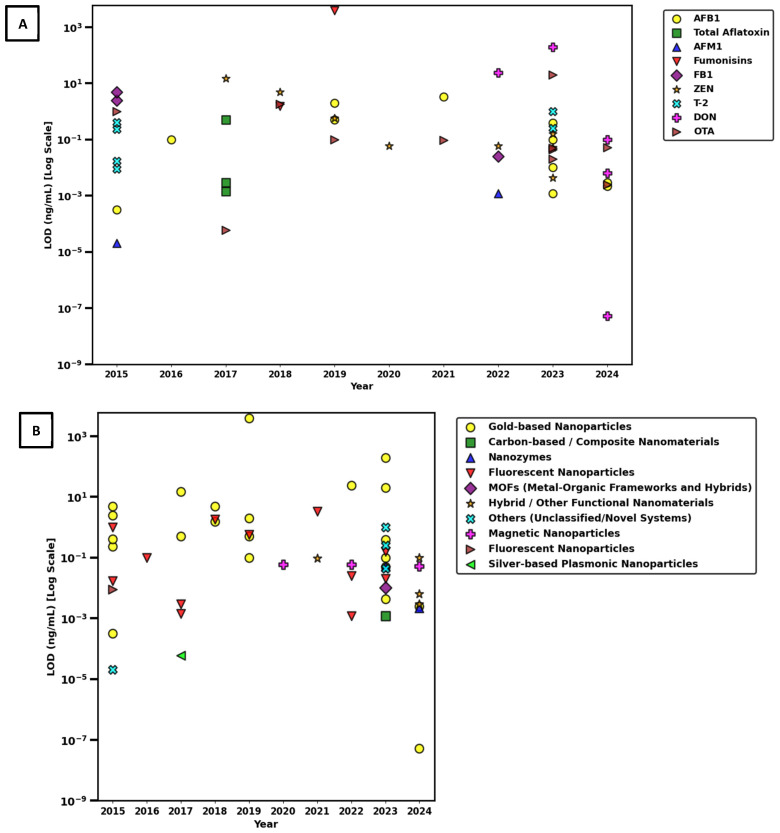
Evolution of LOD for singular mycotoxin detection in nanoparticle-based LFAs, 2015–2024. Panel (**A**) illustrates the limit of detection (LOD, ng/mL on a log scale) reported in singular LFAs for individual mycotoxins across studies published between 2015 and 2024. Each data point corresponds to a specific mycotoxin, including AFB1, total aflatoxins, AFM1, fumonisins, FB1, ZEN, T-2, DON, and OTA, as indicated by distinct markers in the legend. Panel (**B**) presents the same temporal evolution of LOD values, stratified by the type of nanoparticle used in the LFA system (e.g., AuNPs, fluorescent nanoparticles, nanozymes, magnetic nanocomposites, etc.). This comparison allows for assessment of how nanomaterial selection impacts assay sensitivity trends in singular-format LFAs over time.

**Figure 7 toxins-17-00348-f007:**
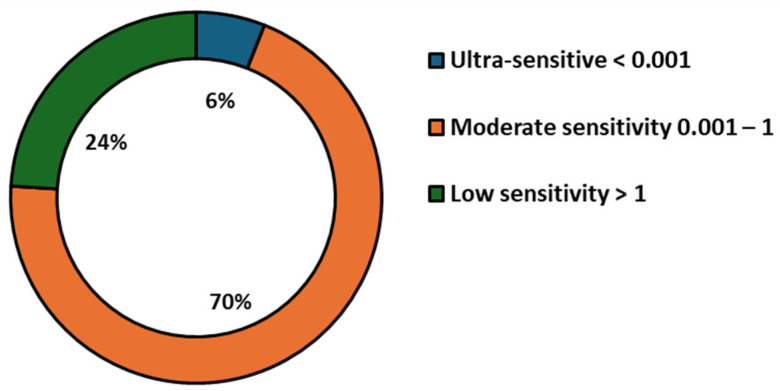
Classification of singular LFA systems based on LOD performance tiers. Only 6.0% of systems achieved ultra-sensitive detection (<0.001 ng/mL), while 70.0% fell within the moderate sensitivity range (0.001–1 ng/mL), and 24.0% were classified as low sensitivity (>1 ng/mL).

**Figure 8 toxins-17-00348-f008:**
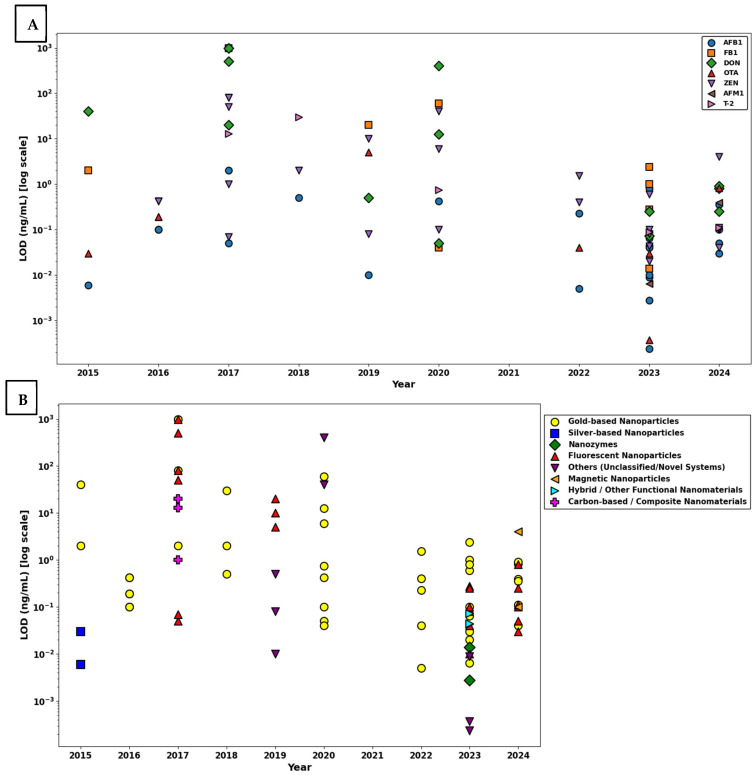
Evolution of LOD for multiplex mycotoxin detection in nanoparticle-based LFAs, 2015–2024. Panel (**A**) displays the LOD, ng/mL on a logarithmic scale for individual mycotoxins detected using multiplex-format LFAs. Each symbol represents a specific target analyte (e.g., AFB1, FB1, DON, OTA, ZEN, AFM1, T-2), enabling comparison of analytical sensitivity across toxins within multiplex detection systems. Panel (**B**) presents the corresponding LOD values stratified by nanoparticle type used in the construction of multiplex LFAs, including gold-based nanoparticles, nanozymes, magnetic nanomaterials, fluorescent nanomaterials, hybrid/functionalized systems, and carbon-based composites. This panel highlights how material innovations contribute to sensitivity trends in multiplex detection platforms. Together, the two panels illustrate both analyte-specific and nanomaterial-specific variation in detection performance and collectively underscore the progressive evolution of nanoparticle-based multiplex LFAs for simultaneous mycotoxin monitoring in food safety applications.

**Figure 9 toxins-17-00348-f009:**
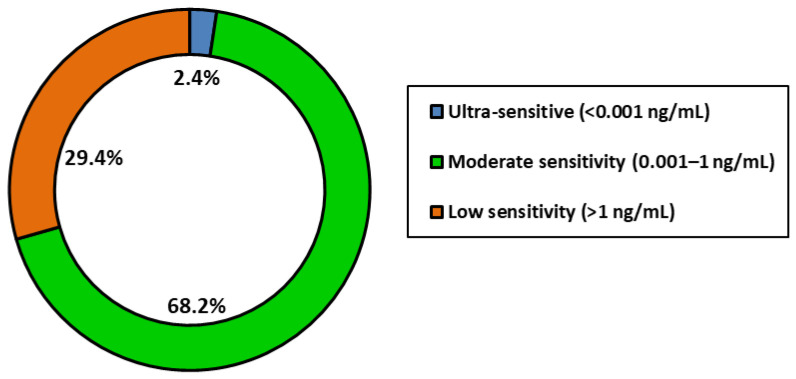
Classification of multiplex LFA systems based on detection sensitivity. Only 2.4% of multiplex LFA systems achieved ultra-sensitive detection (<0.001 ng/mL), while 68.2% demonstrated moderate sensitivity (0.001–1 ng/mL). A total of 29.4% of systems were categorized as having low sensitivity (>1 ng/mL).

**Figure 10 toxins-17-00348-f010:**
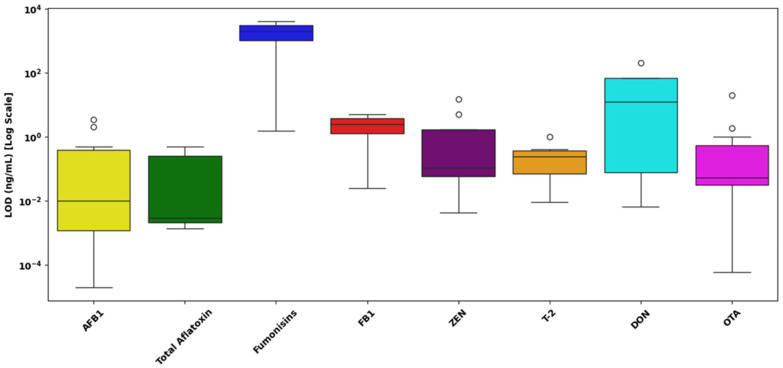
Box plot illustrates the LOD for singular detection of various mycotoxins (AFB1, Total Aflatoxins, Fumonisins, FB1, ZEN, T-2, DON, and OTA) as determined by singular lateral flow assays. LOD values are presented on a logarithmic scale (ng/mL). Each box represents the interquartile range (IQR), with the horizontal line inside indicating the median. Whiskers extend to the minimum and maximum values within 1.5 times the IQR from the lower and upper quartiles, respectively. Data points beyond this range are plotted as individual outliers.

**Figure 11 toxins-17-00348-f011:**
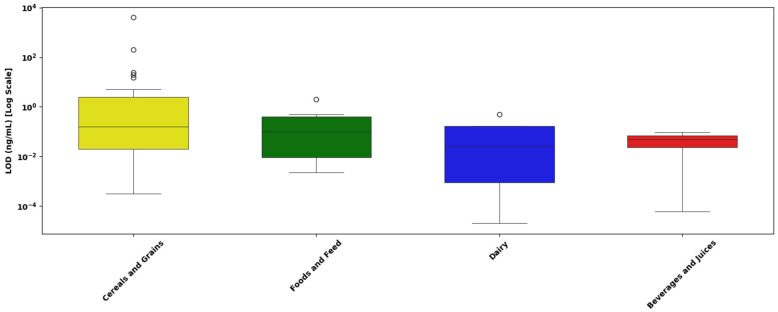
Box plot illustrates the LOD for singular detection of mycotoxins in different sample matrices (Cereals and Grains, Foods and Feed, Dairy, Beverages and Juices), as determined by singular LFAs. LOD values are presented on a logarithmic scale (ng/mL). Each box represents the interquartile range (IQR), with the horizontal line indicating the median. Whiskers extend up to 1.5 times the IQR, and data points beyond this threshold are plotted as individual outliers.

**Figure 12 toxins-17-00348-f012:**
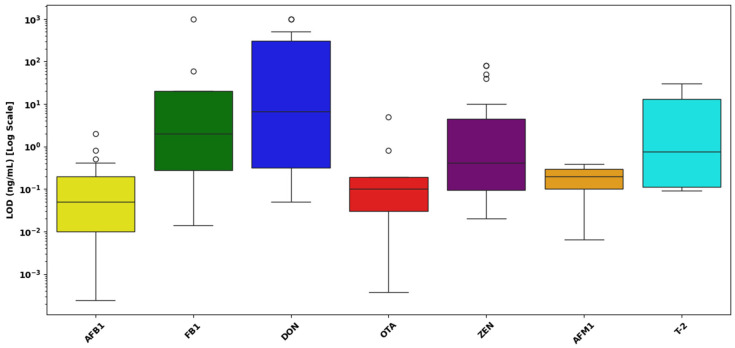
Box plot illustrates the LOD for various mycotoxins as determined by multiplex LFAs. LOD values are presented on a logarithmic scale (ng/mL) to accommodate the wide dynamic range across different mycotoxins. Each box represents the interquartile range (IQR), with the horizontal line inside indicating the median. Whiskers extend to the minimum and maximum values within 1.5 times the IQR from the lower and upper quartiles, respectively. Data points beyond this range are plotted as individual outliers.

**Figure 13 toxins-17-00348-f013:**
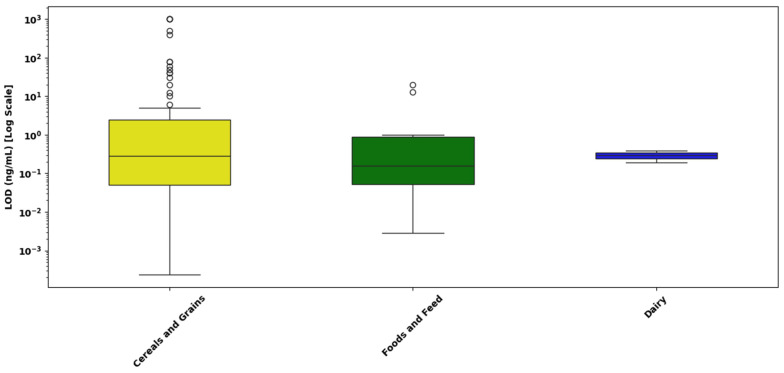
Box plot illustrates the LOD for mycotoxins across different sample matrices cereals and grains, foods and feed, and dairy as determined by multiplex LFAs. LOD values are displayed on a logarithmic scale (ng/mL). Each box represents the interquartile range (IQR), with the horizontal line indicating the median. Whiskers extend to the minimum and maximum values within 1.5 times the IQR, and data points beyond this threshold are plotted as individual outliers.

**Table 1 toxins-17-00348-t001:** Summary of nanoparticle-based lateral flow assays for mycotoxin detection: detection methods, sample types, nanoparticle technologies, sensitivity, and advancements (2015–2024).

Detection Method	Sample	Target Analyte	Nanoparticle	Advancement	Sensitivity	Year	Reference
Individual detection	Rice extract	AFB1	Blue gold nanoflowers (AuNFs)	75 ± 5 nm	0.32 pg/mL	2015	[122]
	Suspicious Fungi-Contaminated Food Samples	AFB1	AuNPs with BSA and AFB1 antibody	Rapid and sensitive AuNP immunochromatographic strip	2 ng/mL	2019	[123]
	Maize	AFB1	Zn-CN (pyrolyzed ZIF-8 metal–carbon nanomaterial)	Colorimetric, fluorescent, and photothermal detection	0.0012 ng/mL (Colorimetric), 0.0094 ng/mL (Fluorescent), 0.252 ng/mL (Photothermal)	2023	[124]
	Various Food Samples	AFB1	AuNPs@SH-poly A-cDNA	Turn on mode-based aptamer sensor AuNPs@SH-poly A-cDNA nanoprobes	0.1 μg/kg	2023	[125]
	Food samples	AFB1	CuCo@PDA nanozyme (Copper-Cobalt polydopamine nanozyme)	Dual-readout (naked eye/smartphone), catalytic colorimetric amplification using peroxidase-like nanozyme	2.2 pg/mL	2024	[126]
	Distillers’ grains	AFB1	Red Fluorescent Microspheres	Enables rapid, sensitive, and accurate on-site detection with cost-effective, large-scale applicability.	3.4 μg/kg	2021	[127]
	Potable Water Samples	AFB1	AuNPs	Lateral flow immunostrip effective for 3 months at 4 °C.	0.5 ppb	2019	[128]
	Food and Feed	AFB1	AuNPs	Dot-blot assay using octapeptide-conjugated AuNPs	0.39 μg/kg	2023	[129]
	Soybean Sauce	AFB1	Eu-nanospheres	Time-resolved fluorescence immunochromatography	0.1 µg·kg^−1^	2016	[130]
	Rice and Peanut	Total Aflatoxin	Quantum Dot Nanobeads (QDNBs)	On-site, ultra-sensitive, and quantitative test strip rapid detection with high sensitivity.	1.4 pg/mL (rice), 2.9 pg/mL (peanut)	2017	[131]
	Maize, Lotus seed	AFB1	Polydopamine-coated HKUST MOFs (HKUST@PDA)	HKUST@PDA as a signal amplification marker.	0.01 ng/mL	2023	[132]
	Maize, Lotus seed	AFB1	UiOL@AIEgens nanocomposites (MOF with AIEgens)	Dual-modal (visual + quantitative), signal-enhanced detection using MOF-AIEgens hybrid	0.003 ng/mL	2024	[133]
	Soy-based foods (soy protein, soy milk)	Aflatoxins (B1, M1, G1, G2, B2)	AuNPs	Monoclonal antibody (3B6) for rapid detection of multiple aflatoxins	0.5 μg/kg	2017	[134]
	Raw milk	AFM1	Immunomagnetic nanobeads	Two types of IMNBs to enhance sensitivity and eliminate sample pretreatment.	0.02 μg/L	2015	[135]
	Milk	AFM1	Fluorescent Nanocomposites	Ratiometric hue-based visual & quantitative detection	0.0012 ng/mL	2022	[136]
	Maize flour	Fumonisins	CdSe/ZnS, QDs + AgNPs + AuNPs	Fluorescence quenching and recovery to enhance sensitivity	1.56 ng/mL	2018	[137]
	Corn samples	FB1	Colloidal AuNP	Rapid, specific, and low-cost immunoassay	2.5 ng/mL	2015	[138]
	Grains	FB1	Urchin-like AuNPs	ICS with UGNs for enhanced sensitivity and rapid detection	5 ng/mL	2015	[139]
	Corn	FB1	Europium (Eu) Nanoparticles	Indirect signal amplification, higher sensitivity	0.025 ng/mL	2022	[140]
	Maize grains	Fumonisins	AuNPs	Optimized IgY-based assay with improved specificity and sensitivity	4000 µg/kg	2019	[141]
	Corn	ZEN	Pt@AuNF nanozyme and horseradish peroxidase (HRP)	Dual enzyme catalytic signal amplification strategy	0.052 ng/mL	2023	[142]
	Cereals	ZEN	AuNPs loaded black phosphorus (BP-Au) nanocomposite	Photothermal LFIA, high sensitivity, excellent photothermal conversion efficiency, and effective on-site monitoring.	4.3 pg/mL	2023	[143]
	Cereals	ZEN	Carboxyl group-coated Fe_3_O_4_ nanoparticles (MNPs)	Portable, dual detection mode, multi-channel ICA with a smartphone-based readout device, Magnetic enrichment for improved sensitivity and robustness	0.06 μg kg^−1^	2020	[144]
	Wheat	ZEN	Colloidal AuNP (30 nm)	Rapid ICS test; optimized antigen and antibody concentrations; completed in 5 min, Millipore 135 NC membrane	15 ng/mL	2017	[145]
	Cereals	ZEN	30% Lu^3+^-doped UCNPs	Novel UCNPs-ICA with optimized optical properties; high specificity; quick detection	0.16 μg/kg	2023	[146]
	Cereals	ZEN	Prussian Blue Nanoparticles (PBNPs)	Portable smartphone-based readout with quantitative analysis	0.12 μg/kg	2022	[147]
	Corn	ZEN	AuNPs	Rapid detection in 5 min, competitive assay with aptamer and complementary sequence	5–200 ng/mL	2018	[148]
	Corn	ZEN	QDs	Fluorescent quenching lateral flow assay	0.58 ng/mL	2019	[149]
	Cereals	T-2	AuNP + selenium nanoparticles (SeNPs)	SeNPs (Se-ICS) and dual AuNPs (Duo-ICS) for improved sensitivity	Duo-ICS: 1 ng/mL; Se-ICS: 0.25 ng/mL	2023	[150]
	Rice, chicken feed	T-2	Colloidal Gold (CG) and Fluorescent Microspheres (FMs)	Comparison of CG and FMs as labels in LFIA, with optimized cut-off values	0.23 μg/kg (rice), 0.41 μg/kg (chicken feed)	2015	[151]
	Rice, maize, feed	T-2	Eu(III) nanoparticles	Time-resolved fluorescence for ultrasensitive detection	Rice 0.09 ng/g Feed 0.17 ng/g	2015	[152]
	Food samples	DON	Cauliflower-like ReS_2_@Pt core–shell nanospheres	Colorimetric-catalytic dual-mode, peroxidase-mimicking nanozyme with enhanced antibody affinity	6.5 pg/mL	2024	[153]
	Corn, wheat, naturally contaminated cereals and feed	DON	Core–shell up-conversion nanoparticles	Enhanced up-conversion luminescence for highly sensitive and specific DON detection within 5 min	0.1 ng/mL	2024	[154]
	Wheat	DON	AuNPs	Single strip with three test lines (TTLS) for semi-quantitative and quantitative determination	200 µg/kg	2023	[155]
	Corn	DON	AuNPs	Aptamer-based lateral flow assay	24.11 ng/mL	2022	[156]
	Grain	DON	AuNR@Ag@SiO2-AuNP core–shell-satellite nanoassembly	Highly SERS-active, stable, antibody-modified core–shell-satellite structure	0.053 fg/mL	2024	[157]
	Real maize samples	OTA	Ultrabright green-emissive AIE nanoparticles (AIENPs)	Enhanced detectability of LFIA with ultrabright AIENPs; applicability for small molecules and macromolecules	0.043 ng/mL	2023	[158]
	Wine, beer, apple juice, milk samples	OTA	Aptamer-conjugated AuNPs	Aptasensor strip for rapid detection; competitive format; visual and semi-quantitative detection	Visual LOD: 0.05 ng/mL Semi-quantitative LOD: 0.02 ng/mL	2023	[159]
	Coffee samples	OTA	AuNPs	Barcode-style lateral flow assay for semi-quantitative detection; distinct color patterns for different OTA concentrations	2.5 µg/L	2024	[160]
	Maize and grape juice	OTA	AuNP nanobipyramids	Photothermal immunoassay with Alkaline phosphatase-mediated in situ growth of AuNBPs; sensitive detection using a thermometer	020 ng/mL	2023	[161]
	*Astragalus membranaceus*	OTA	Aptamer-modified MNPs	Three-in-one lateral flow aptasensor using aptamer-MNPs for purification, enrichment, and detection	0.053 ng/mL	2024	[162]
	Grape juice	OTA	Magneto-gold nanohybrid (MGNH)	Novel MGNH integrated into LFIA for simultaneous magnetic separation and colorimetric target sensing	0.094 ng mL^−1^	2021	[163]
	Wheat, beer	OTA	Ytterbium-doped sodium yttrium fluoride (NaYF_4_:Yb,Er) UCNPs	Aptamer-based upconversion fluorescent strip	1.86 ng/mL	2018	[164]
	Grape Juice, Wine	OTA	Silver nanoparticles	Silver nanoparticle-based fluorescence-quenching lateral flow immunoassay	0.06 µg/L	2017	[165]
	Wheat, Maize, Soybean, Rice	OTA	Fluorescent europium (III) [Eu (III)] nanoparticles (EuNPs)	Time-resolved fluorescent ICA	1.0 μg kg^−1^	2015	[166]
	Rice, Corn, Ginger, Green Bean	OTA	Microorganism-loaded AuNPs	Use of Yeast/Lactobacillus as reducers and carriers for AuNP synthesis, enhancing adsorption and lowering antibody usage	0.1 ng/mL	2019	[167]
	Wheat	OTA	Europium nanospheres	Smartphone-enabled iPOCT for rapid detection; fluorescent lateral flow assay; cloud-based result sharing	0.02 ng/mL	2023	[168]
Multiplexing	Maize Flour	AFB1, FB1	Desert rose-like gold nanoparticles (DR-GNPs), Red spherical GNPs	Multicolor ICST strip test employing DR-GNPs and red spherical GNPs	2 μg/kg (AFB1) 1000 μg/kg (FB1)	2019	[169]
	Maize	FB1, DON	AuNPs	Silver staining for signal amplification	2.0 ng/mL (FB1), 40 ng/mL (DON)	2015	[170]
	Maize meal	AFB1, OTA	Ag@Au Core–Shell NPs	SERS labels embedded Ag@Au core–shell NPs for sensitive double detection without nucleic acid amplification	0.006 ng/mL (AFB1) 0.03 ng/mL (OTA)	2015	[171]
	Food sample	AFB1, FB1	Fe-N-C single-atom nanozymes (SAzymes)	Ultra-sensitive detection, dual-function label & catalyst, smartphone readout	2.8 pg/mL AFB1), 13.9 pg/mL (FB1)	2023	[172]
	Corn, Rice, Peanut	AFB1, ZEN, OTA	AuNPs	Systematic optimization of antibody-AuNP conjugates, nanoparticle size, and capture antigen position for improved detection	0.10–0.13 μg/kg (AFB1), 0.42–0.46 μg/kg (ZEN), 0.19–0.24 μg/kg (OTA)	2016	[173]
	Milk, Maize and Wheat	AFB1, AFM1, OTA	AuNP iridium nanozyme	Three-channel aptamer-based lateral flow assay (Apt-LFA), Catalytic chromogenic substrate & fluorescence-based optimization	0.39 ng/mL (AFM1), 0.36 ng/mL (AFM1), 0.82 ng/mL (OTA)	2024	[174]
	Maize	AFB1, OTA, ZEN	Flower-like AuNPs and red-emitting quantum dots	Multiplexed competitive lateral flow immunoassay (cLFIA) based on inner filter effect (IFE)	0.005 μg/L (AFB1), 0.04 μg/L (OTA), 0.4 μg/L (ZEN)	2022	[175]
	Maize- and cereal-based animal feeds	AFB1, ZEN, T-2	AuNPs	Multi-color nanoparticles in an immunochromatographic strip for the simultaneous detection	0.5 ng/Ml (AFB1), 2 ng/mL (ZEN), 30 ng/mL (T-2)	2018	[176]
	Corn, Wheat	AFB1, DON, ZEN	Core–shell up-conversion nanoparticle	Smartphone-integrated UCNPs for portable, simultaneous multi-mycotoxin detection	DON: 0.25 ng/mL, AFB1: 0.05 ng/mL, ZEN: 0.1 ng/mL	2024	[177]
	Wheat, Corn, Animal Feed	AFB1, DON, ZEN	Carboxylated latex nanospheres with phycocyanin and mAbs	Fluorescent multiplex detection using Phycocyanin-labeled LNS, visualized via LED UV and smartphone in <25 min	AFB1 Wheat: 1.04 ng/mL Corn: 1.6 ng/mL Feed: 2.08 ng/mL DON Wheat: 2.2 ng/mL Corn: 6.45 ng/mL Feed: 2.9 ng/mL ZEN Wheat: 1.74 ng/mL Corn: 1.67 ng/mL Feed: 2.11 ng/mL	2024	[178]
	Grains	AFB1, ZEN, DON	Carboxylated Eu(III)-chelate-doped polystyrene nanobeads	Time-resolved fluorescence to reduce background noise; quantitative multiplex detection with portable reader	AFB1: 0.03 ng/g, ZEN: 0.11 ng/g, DON: 0.81 ng/g	2024	[179]
	Corn, Rice, Wheat	AFB1, OTA	Au@SiO_2_ SERS nanotags	Multiplex and ultrasensitive detection using SERS-based LFIA; high sensitivity and biocompatibility	AFB1: 0.24 pg/mL, OTA: 0.37 pg/mL	2023	[180]
	Corn	AFB1, ZEN	Magnetic Fe_3_O_4_@PEI/AuMBA@Ag-MBA nanocomposites	Bi-channel SERS-based LFIA strip for simultaneous detection, magnetic enrichment improves sensitivity	0.1–10 μg/kg (AFB1) 4–400 μg/kg (ZEN)	2024	[181]
	Wheat & Corn	FB1, DON, ZEN	AuNPs	Multiplex qualitative detection, High specificity	60 ng/mL (FB1), 12.5 ng/mL (DON), 6 ng/mL (ZEN)	2020	[182]
	Cereal	OTA, AFB1, FB1, ZEN	QDs, AuNPs, Dendritic mesoporous silica nanoparticles (DMSNs)	Homogeneous fluorescence immunoassay for simultaneous detection of four mycotoxins	0.0001 μg/L (OTA) 0.0008 μg/L (AFB1) 0.001 μg/L (FB1) 0.0006 μg/L(ZEN)	2023	[183]
	Cereal	ZEN, FB1, OTA, AFB1	AuNPs with fluorophore-labeled ssDNA	Simultaneous quantitative detection of four mycotoxins using a single test process with fluorescence recovery	0.02 μg/kg (ZEN) 2.42 μg/kg (FB1) 0.03 μg/kg (OTA) 0.065 μg/kg (AFB1)	2023	[184]
	Corn	ZEN, OTA, FB1	Tricolor QBs	Simultaneous qualitative detection of multiple mycotoxins	5 ng/mL (OTA), 20 ng/mL (FB1), 10 ng/mL (ZEN)	2019	[185]
	Environmental water	Pesticide residues (imidacloprid, pyraclostrobin) and mycotoxin (AFB1)	SERS nanotags	Dosage-sensitive and simultaneous quantitative SERS-based LFIA for multiple pollutants	8.6 pg/Ml (imidacloprid) 97.4 pg/mL (pyraclostrobin) 8.9 pg/mL (AFB1)	2023	[186]
	Foods	DON, AFB1, ZEN	Cu_2_-xSe-Au nanocomposites	Multi-target photothermal immunochromatography for simultaneous detection	73 ng/L (DON) 45 ng/L (AFB1) 43 ng/L (ZEN)	2023	[187]
	Maize	DON, T-2, ZEN	Amorphous carbon nanoparticles	Utilization of amorphous carbon nanoparticles (ACNPs) as detection labels	20 μg/kg (DON), 13 μg/kg (T-2), 1 μg/kg (ZEN)	2017	[188]
	Feedstuff, naturally contaminated	AFB1, ZEN, DON	CdSe/SiO_2_ QBs	Simultaneous determination, Quick analysis.	10 pg mL^−1^ (AFB1) 80 pg mL^−1^ (ZEN) 500 pgmL^−1^ (DON)	2019	[189]
	Cereals and feed (naturally contaminated)	AFB1, FB1, DON, T-2, ZEN	UiO-66-NH_2_@quantum dot (NU66@QD) nanocomposites	High bio affinity and controllable assembly nanocarrier for simultaneous detection of five mycotoxins	0.04 μg/kg (AFB1) 0.28 μg/kg (FB1) 0.25 μg/kg (DON) 0.09 μg/kg (T-2) 0.08 μg/kg (ZEN)	2023	[190]
	Maize	AFB1, ZEN	Lu^3+^-doped UCNPs	Synthesis and functionalization of Lu^3+^-doped UCNPs with larger size, more regular structure, and significantly brighter fluorescence intensity	LOD: 0.01 ng/mL AFB1, LOD: 0.1 ng/mL ZEN	2023	[191]
	Maize and its products	AFB1, ZEN	Eu/Tb(III) nanospheres	Time-resolved fluorescence immunochromatographic assay (TRFICA) using anti-idiotypic nanobody (AIdnb) and monoclonal antibody (mAb)	Aflatoxin B1 (AFB1): 0.05 ng·mL^−1^ Zearalenone (ZEN): 0.07 ng·mL^−1^	2017	[192]
	Food and feed	AFB1, ZEN	AuNPs	Dual lateral flow immunochromatographic assay for simultaneous detection	0.23 μg/L (AFB1) 1.53 μg/L (ZEN)	2022	[193]
	Soybean, corn, rice	AFB1, OTA	Quantum dot nanobeads	Bispecific monoclonal antibody-based multiplex LFIA	0.037 μg/kg (AFB1), 1.19 μg/kg (OTA)	2016	[194]
	Milk	Melamine (MEL), Enrofloxacin (ENR), Sulfamethazine (SMZ), Tetracycline (TC), AFM1	AuNPs	Multiple lateral flow immunoassay (LFIA) for simultaneous detection of 5 chemical contaminants	0.173 ng/mL (MEL) 0.078 ng/mL(ENR) 0.059 ng/mL (SMZ) 0.082 ng/mL(TC), 0.0064 ng/mL (AFM1)	2023	[195]
	Maize and wheat samples, naturally contaminated	ZEN, DON	CdSe/CdS & CdSe/CdS/ZnS core–shell heterostructures	Multicolor lateral flow immunoassay using QD bioconjugates	40 μg kg^−1^ (ZEN), 400 μg kg^−1^ (DON)	2020	[196]
	Maize, Wheat	ZEN DON	Indium Phosphide (InP) QDs	Water-soluble InP/ZnS QDs-based fluorescent nanostructures (QD@SiO_2_) for simultaneous detection	50 µg/kg (ZEN) 500 µg/kg (DON)	2017	[197]
	Wheat	DON, ZEN, T2/HT2	CdSe/ZnS QDs, Colloidal gold (CG)	QD-based LFIA consumed less immunoreagents, more sensitive, lower false negative rate; CG-based LFIA developed for comparison	1000 μg/kg (DON), 80 μg/kg (ZEN), 80 μg/kg (T2/HT2)	2017	[198]
	Cereals	AFB1, ZEN, DON, T-2, FB1	AuNPs + Time resolved fluorescence microspheres	Smartphone-based dual detection mode device; multiplex detection; integrated visible light and fluorescence detection	qLODs: 0.59/0.24/0.32/0.9/0.27 μg/kg (AuNPs), 0.42/0.10/0.05/0.75/0.04 μg/kg (TRFMs)	2020	[199]
	Cereals	DON, ZEN, T-2, TEA, AOH (Total: 15 mycotoxins)	AuNPs	Simultaneous detection of 15 mycotoxins in a single test	DON: 0.91, ZEN: 0.04, T-2: 0.11, TEA: 0.12, AOH: 0.09–0.	2024	[200]

## Data Availability

No new data were created or analyzed in this study.

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
