# Peer review of "Recent Advancements in Lateral Flow Assays for Food Mycotoxin Detection: A Review of Nanoparticle-Based Methods and Innovations"

_toxins, 2025, doi:10.3390/toxins17070348_

Round 1

Reviewer 1 Report

Comments and Suggestions for Authors

This review article provides a comprehensive overview of recent advancements (2015-2024) in nanoparticle-based lateral flow assays (LFAs) for food mycotoxin detection, featuring thorough analysis and discussion. The authors systematically cover the application of various nanomaterials (gold nanoparticles, fluorescent nanoparticles, magnetic nanoparticles, etc.) in LFAs and analyze their impact on detection performance. Data extracted from 78 relevant studies is systematically analyzed, with trends and variations in LFA performance clearly presented using figures and statistical methods. The article offers an in-depth exploration of the challenges and opportunities for LFA technology in food mycotoxin detection and proposes future directions and recommendations. In my opinion, this manuscript is suitable and qualified to be published in toxins after minor revision according to the comments below.

  1. The legend for Figure 5 is insufficiently clear. The labeling of "Panel A" and "Panel B" in the text does not clearly correspond to their content in the figure (e.g., which panel shows data stratified by mycotoxin type vs. nanoparticle type). A similar lack of clarity exists for Figure 7.
  2. The page numbering is discontinuous. For instance, the number "379" appears abruptly at the end of page 17.
  3. Data Accuracy (Page 19): The stated LOD for DON using the SERS platform is reported as "5.3×10⁻¹¹ng/mL" (citing reference [157]). However, reference [157] (2024) actually reports an LOD of 0.053 fg/mL. This data point must be corrected.
  4. The introduction could be more concise when covering the hazards of food mycotoxins and the advantages of LFA technology, avoiding excessive repetition of well-established information.
  5. The conclusion's proposal to "move beyond a material-centric focus" is insightful. However, this point would be strengthened by suggesting specific areas for optimization, such as improving antibody affinity or membrane flow dynamics.
  6. Reference [128] cited on page 36 (regarding AFB1 LOD) is incorrectly listed as 2019.Reference [163], described on page 36 as a 2024 study involving aptamers ("aptamer-based three-channel assay"), is actually from 2021 and does not involve aptamer technology at all. This description and the citation year are incorrect.
Comments on the Quality of English Language

There are some typographic formats or grammatical errors, please check the paper throughout carefully, such as: in Table 1, the word “fuminosins” is incorrect.

Reviewer 2 Report

Comments and Suggestions for Authors

Dear Authors

The article is well written and comprehensive. This study synthesizes developments in mycotoxin detection using nanoparticle-based lateral flow assays (LFAs) in a timely, thorough, and methodologically sound manner (2015–2024). The study offers important insights into material advancements, performance trends, and regulatory alignment while addressing a pressing demand for quick, on-site food safety monitoring solutions. Particularly reliable is the statistical examination of sensitivity trends (e.g., multiplex vs. single tests).

I have very minor comments

Do the commercial sources of nanoparticle-based LFAs have an impact on test effectiveness and sensitivity?

Line 1117-118: do you mean with “masa production”, do you mean mass

Line 158: “mycotoxin detection [53-57]” should be followed by full stop

Line 1034: The reference “Shahjahan et al. (2023)” should be replaced by numbering style

Reviewer 3 Report

Comments and Suggestions for Authors

The article is generally well written and fits the scope of the journal. The narrative is well thought and an important topic is reviewed.

 I would recommend the following points to be addressed:

  1. Line 189-194 the units between different essays is discussed. This discussion is important and it should be further developed to answer which of these limits tends to underestimate the real concentration? Most samples are solid so I assume that the conversion form solid to liquid dilutes the sample and hence the liquid concentration is an overestimation? Is this the case?
  2. In regard to Figures a general Figure displaying the functioning principle of the lateral flow assay works would be necessary.
  3. In regard to Figures. Figures are generally well chosen in the context of this work but some details are missing when it comes to labelling and reading the figures. Examples: . 2) Figure 2. Magnetic particles (red hexagons) and green triangles are not labelled. 1) Figure 1A – antibodies are only labelled at the bottom of the figure (orange blue sticks) but they appear earlier and should be labelled as they appear
  4. Figures 9,10,11,12 where average and errors are shown are given on a log scales. This makes it very hard to read. I would suggest transforming this to linear scale. Cuts in the axes are better. Previous figures make sense with a log scale since the data are very scattered but not the averages.

Some references are misplaced. Please check ALL.

  1. Line 1090…’’More notably, in the domain of 1090 fumonisin detection, a 2017 multicolor ICST multiplex assay targeting FB1 showed an LOD of 1000µg/kg [149]’’

Reference 149 is: 149. Chen, Y.; Fu, Q.; Xie, J.; Wang, H.; Tang, Y. Development of a High Sensitivity Quantum Dot-Based Fluorescent Quenching 1510 Lateral Flow Assay for the Detection of Zearalenone. Anal. Bioanal. Chem. 2019, 411, 2169–2175. https://doi.org/10.1007/s00216-1511 019-01652-1

The reference is not corelated with the text 2017/2019 and fumonisin and Zearalenone is not really the same thing.

  1. ‘’ Meanwhile, a 2019 study using AuNPs 1094 in maize flour reported a fumonisin LOD of 4000µg/kg [128]’’

[128]. Yan, X.; Persaud, K.C. The Optimization of a Lateral Flow Immunoassay for Detection of Aflatoxin B 1 in Potable Water Samples. 1452 IEEE Sens. J. 2018, 19(2), 404-412. 10.1109/JSEN.2018.2878449

AGAIN the reference does not corelate with the text. The cited study is 2018 in the first place and also not about maze.
